# A model of head direction and landmark coding in complex environments

Yijia Yan[1,2]*, Neil Burgess[1], Andrej Bicanski[1,3]*

**1** Institute of Cognitive Neuroscience, University College London, London, United Kingdom, **2** Nuffield Department of Clinical Neurosciences, University of Oxford, Oxford, United Kingdom, **3** School of Psychology, Newcastle University, Newcastle upon Tyne, United Kingdom

* yijia.yan@ndcn.ox.ac.uk (YY); andrej.bicanski@gmail.com (AB)

**Data Availability Statement:** Animal behavioural data are within the Supporting Information files (S1 Dataset). Simulation results data are available on Kaggle: www.kaggle.com/chronowanderer/ aLBcells-for-HDsystem-simulation-results Relevant

## Abstract

Environmental information is required to stabilize estimates of head direction (HD) based on angular path integration. However, it is unclear how this happens in real-world (visually complex) environments. We present a computational model of how visual feedback can stabilize HD information in environments that contain multiple cues of varying stability and directional specificity. We show how combinations of feature-specific visual inputs can generate a stable unimodal landmark bearing signal, even in the presence of multiple cues and ambiguous directional specificity. This signal is associated with the retrosplenial HD signal (inherited from thalamic HD cells) and conveys feedback to the subcortical HD circuitry. The model predicts neurons with a unimodal encoding of the egocentric orientation of the array of landmarks, rather than any one particular landmark. The relationship between these abstract landmark bearing neurons and head direction cells is reminiscent of the relationship between place cells and grid cells. Their unimodal encoding is formed from visual inputs via a modified version of Oja's Subspace Algorithm. The rule allows the landmark bearing signal to disconnect from directionally unstable or ephemeral cues, incorporate newly added stable cues, support orientation across many different environments (high memory capacity), and is consistent with recent empirical findings on bidirectional HD firing reported in the retrosplenial cortex. Our account of visual feedback for HD stabilization provides a novel perspective on neural mechanisms of spatial navigation within richer sensory environments, and makes experimentally testable predictions.

## Author summary

Animals need a 'compass' to maintain a sense of orientation relative to their environment in order to support spatial navigation and memory. This 'compass' has been identified with head direction cells, found in various species. These cells fire at a high rate when an animals' head is facing in a specific allocentric direction regardless of the current location. Their firing needs to be stabilized by sensory inputs to prevent drift. Since the discovery of head direction cells most experiments and computational models have employed visual input in the form of a single polarizing cue. It is unknown how the global sense of

computational modelling codes (in MATLAB 2017a) are available on GitHub: www.github.com/Chronowanderer/aLBcells-for-HDsystem and www.github.com/Bicanski/aLBcells-for-HDsystem.

**Funding:** AB and NB acknowledge funding by the European Research Council (erc.europa.eu) No. 694779, the Wellcome Trust (wellcome.org) No. 202805/Z/16/Z, and Horizon 2020 (ec.europa.eu/programmes/horizon2020) No. 785907. The funders had no role in study design, data collection and analysis, decision to publish, or preparation of the manuscript.

**Competing interests:** The authors have declared that no competing interests exist.

direction can be maintained in more complex environments with multiple cues of differing stability, salience, and directional specificity. Our model suggests that (contrary to models with simple cues) a new type of neuron (abstract landmark bearing cells) endows the head direction system with powerful abilities, including unimodal landmark encoding at a sensory level despite partially conflicting cues, robustness against unreliable and ephemeral cues, and a high encoding capacity across environments. The model is consistent with numerous empirical findings, and provides a novel perspective on the neural mechanisms of spatial navigation in more realistic, cue-rich settings across multiple environments.

## Introduction

The global sense of direction is believed to be supported by head direction (HD) cells. First discovered in rodents, a HD neuron only fires at a high and steady rate when an animal's head is facing in a specific allocentric direction regardless of the current location [1,2]. HD signals have been found in various brain areas in several organisms [3,4], and data consistent with signatures of HD coding have been found in humans [5–8].

HD cells are found along the diencephalic-cingulate-parahippocampal Papez' circuit [9], including the retrosplenial cortex (RSC) [10], dorsal tegmental nucleus of Gudden [11], lateral mammillary nucleus [12,13], anterodorsal thalamic nucleus [14], dorsal presubiculum [1], and medial entorhinal cortex [15]. HD signals were also found in other cortical regions, including the medial parietal cortex in humans [5,16], as well as medial prestriate cortex (or visual association cortex) [17] and deep layers of the primary visual cortex (V1) [18].

HD cells are controlled by both external landmarks and self-motion information [19–21]. The self-motion information includes angular path integration in the vestibular system, which generates unimodal HD signals even before eye-opening [3,22,23]. Path integration via vestibular inputs can maintain HD in darkness, yet is subject to accumulated path integration error resulting in drift in the absence of sensory reset [24].

Anatomical considerations suggest that distinct sub-regions in RSC, namely dysgranular RSC (dRSC) and granular RSC (gRSC), map onto different aspects of information processing, in which dRSC shares more reciprocal connections with visual regions while gRSC is more densely connected with subcortical regions of the HD system [25,26]. HD cells with both directional and direction-by-location firing characteristics have been reported in RSC [10], which has been proposed to be involved in learning external landmark information and location-specific feedback [27], transmitting cortical visual information to subcortical HD circuits [17]. Consistent with this, lesions to dRSC do not disrupt the generation of HD signals but destabilize HD signals across time [3].

More recently, Jacob et al. [28] report so-called bidirectional (BD) HD cells in dRSC. These cells exhibit a bimodal activity profile, firing for two directions across two connected environments that contain identical visual landmarks in opposite directions (180˚ mirrored environments). See Table 1 for all abbreviations used in this paper. HD signals in gRSC and dorsal presubiculum remain unimodal across such conflicting (at the sensory level) environments [28]. This study corroborates the notion that dRSC acts as an integrator of self-motion and visual information, thus providing integrated signals for correcting accumulated path integration error [3,22,23].

However, several crucial questions regarding the integration of visual information with HD signals remain unanswered. How does the HD system extract useful information from

**Table 1. The ascending sorted acronyms.**

| Acronyms | Descriptions |
| --- | --- |
| aLB | Abstract landmark-bearing |
| BC | Between-compartment |
| BD | Bidirectional |
| dRSC | Dysgranular retrosplenial cortex |
| gRSC | Granular retrosplenial cortex |
| HD | Head direction / head-directional |
| IoU | Intersection over Union |
| mOSA | Modified Oja's Subspace Algorithm |
| OSA | Oja's Subspace Algorithm |
| RSC | Retrosplenial cortex |
| V1 | Primary visual cortex |
| Vis | Visual (or V1) |
| WC | Within-compartment |

Acronyms that only occur in figures are not shown in this table, as they are explained in all related captions.

complex visual inputs to enable HD cells to maintain a unimodal representation across the whole environment? Almost all HD modelling studies simply assume that an isolated Gaussian activity packet, represented in visual areas, codes for a single polarizing cue in the environment, e.g. a stereotypical white cue card on the wall of a recording arena. However, although useful for stabilizing HD, it is unclear how such a unimodal Gaussian representation is derived from more complex environments with multiple cues of differing stability, salience, and directional specificity. Considering that real animals likely visit multiple such complex environments also raises the question of capacity: If visual feedback connections allow HD to be reset upon re-entry to a familiar environment, what might the capacity of the system be, and how does the HD system deal with unreliable or ephemeral visual information in rich real-world environments? Here the term 'capacity' refers to the number of distinct environments that can be stored and recalled (i.e. re-instating the appropriate HD) in the future. The present model investigates these questions in order to identify new properties of the HD-visual circuitry, including retrosplenial cortex (RSC).

Computational models have only recently begun to explore the integration of visual landmark information with HD circuits in detail [27], though mostly restricted to simple cue representations and a direct V1-RSC connection (e.g. [29–32]). Here we propose a computational model of visual landmark processing in cortical regions, capable of extracting a unimodal orientation signal under panoramic visual landmarks for supporting the HD system in subcortical regions across multiple complex environments. This is achieved via cells downstream of primary visual areas that integrate visual information via a modified version of Oja's learning rule with negative feedback control on input layers of visual cells [33–35]. The model can maintain unimodal landmark encoding at a sensory level despite conflicting visual information; is robust against unreliable and ephemeral cues; and exhibits a high encoding capacity across multiple environments. We also suggest an experimentally practical scheme to test for novel model components, and discuss the utility of such a unimodal encoding for HD retrieval across multiple environments. The resulting account is coherent with numerous empirical findings, and provides a novel perspective on the neural mechanisms of spatial navigation in more realistic, cue-rich settings (with cues of varying directional specificity) across multiple environments.

## Materials and methods

### Model overview

Fig 1A depicts a schematic of the model, containing two pathways, processing visual landmarks (purple) and HD self-motion cue (orange) respectively. The two pathways meet in dRSC (pink) where the integrated visual/self-motion signal is generated and projected back to the HD attractor in order to correct for drift. The subcortical pathway (orange) transmits and maintains global HD. For simplicity, we employ a single ring attractor to generate HD information regulated by HD angular velocity from the vestibular system [36]. Other models have explored more biologically plausible architectures with spiking neural networks (e.g. [27,31,32,37,38]). For the purpose of modelling, the attractor is assumed to substitute not only the entire generative circuitry for HD signals (i.e. dorsal tegmental nucleus of Gudden and lateral mammillary nucleus), but also anterodorsal thalamic nucleus (being the principal source of HD information entering gRSC), and presubiculum being the main target for feedback from dRSC to the attractor (i.e. pink arrow from dRSC to the attractor).

For the cortical pathway (purple), we employ a new algorithm (see below) to learn a sparse, unimodal representation of a set of visual landmarks in complex scenery containing multiple cues/features with differing directional specificity. Fig 1B provides an illustration of feature-specific visual receptive fields of the visual inputs in the model. Red, blue, and green colors stand for different sensory attributes/features (e.g. colors, texture, contrast etc.) encoded by corresponding arrays of neurons with egocentric directional receptive fields (the top row). Conversely, cues with the same 'color' stand for indistinguishable sensory features encoded by the same array of receptive fields. For instance, the firing rates of visual cells for the 'red' feature exhibit a bimodal distribution of activity when two undistinguishable 'red' cues are located in opposing directions. That is, solely on the basis of the 'red' cue the agent would not be able to distinguish two orientations 180˚ apart. Visual cells for the 'green' feature are silent in the example due to the absence of such a feature in the current scene.

Downstream of visual neurons the learning algorithm (see below) generates a representation of what we term 'abstract landmark-bearing' (aLB) cells. Receiving afferent projections from visual model components, these cells sparsely encode an abstract directional representation, i.e. one that does not correspond to the bearing of any one particular cue but rather a global orientation of the entire egocentric scenery (i.e. a conjunction of cues; Fig 1D). For simplicity we treat all cues as distal so that parallax effects are absent. However, the model is in principle compatible with the proposed role of RSC in location-specific feedback by expanding the dRSC population to include place modulated HD cells [27].

aLB cells anchor more complex arrays of sensory inputs, to HD signals, as compared to single isolated cues. Their downstream targets are dRSC cells, suggested to integrate landmark information with global HD signals received from gRSC (which receives global HD signals from the ring attractor). The model leaves open where aLB cells reside, e.g. in retrosplenial cortex or elsewhere (see Discussion). Key model mechanisms are summarized below. More detailed information is available in S1-S4 Appendices. All hyperparameters chosen for simulations described in the results are given in S1 Table.

### Neural dynamics

We employ a firing-rate based neuron model with firing rates normalized to range from 0 to 1. The firing rate $f(t)$ is derived from the activation level $a(t)$ of a single neuron via a modified hyperbolic tangent function. Unless specified otherwise,

$$f(t) = \tanh(\beta(a(t) - \alpha)) \tag{1}$$

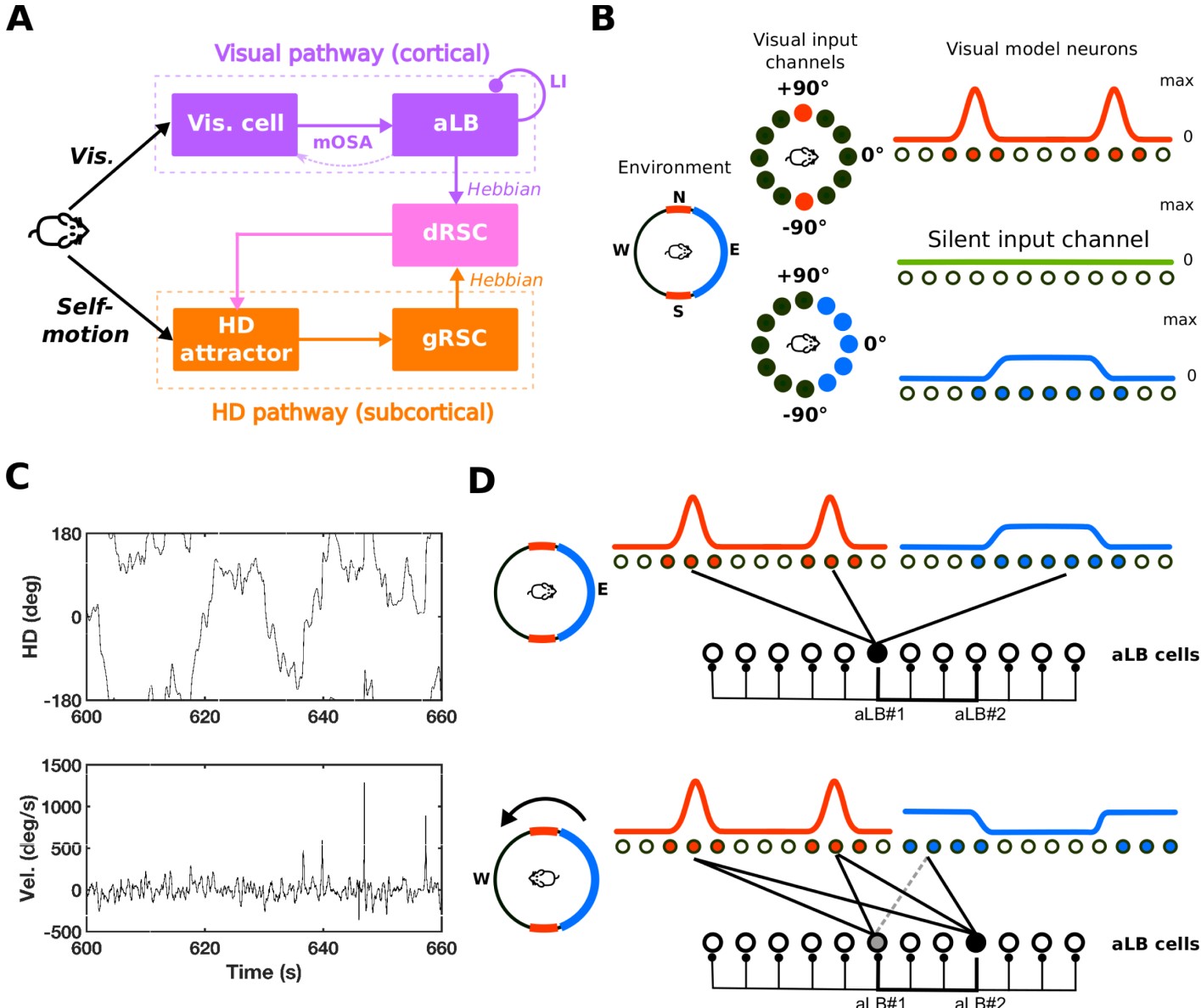

**Fig 1. The basic structure of the HD system with two-stage visual landmark processing.** (A) The systems-level network structure of the HD system which contains two pathways, processing visual landmarks (purple) and HD self-motion inputs (orange) respectively. The two pathways meet in dRSC (pink) where the integrated signal is generated and projected back to the HD attractor in order to correct for drift and keep a global real-time sense of direction. (B) Visual layers consisting of neurons tuned to specific features at specific egocentric directions. Different colors (red, blue, and green) stand for different features. Left: a circular environment; N, E, S, and W represent the allocentric directional frame. Middle: numerical degrees represent the egocentric directional frame (with 0 indicating ahead), here shown for the red and blue feature sensitive neurons. Right: Firing-rate responses across populations of feature-specific neurons. (C) The agent's HD in simulation is taken from data-foraging rat. Top: Allocentric trajectory at the 11th minute of a 20-minute trajectory. Bottom: The corresponding angular velocity at the 11th minute. (D) The unimodal encoding of aLB cells when the agent is facing east (top) and then west (bottom). Two different aLB cells encode sceneries from different facing directions despite a large overlap in perceptual features (red cues). Darker filled circles indicate stronger activation of aLB cells, whilst darker lines indicate stronger feed-forward connections and inhibitory connections among aLB cells. The dashed line (bottom) indicates an example of a connection that has been depressed when the agent was facing East, so that the updated position of the cue in the visual field will not drive a previously active aLB neuron. Abbreviations: HD: head-direction(al); aLB: abstract landmark bearing; dRSC/gRSC: dysgranular/granular retrosplenial cortex; Vis.: visual; mOSA: modified Oja's Subspace Algorithm; LI: lateral self-inhibition; Vel.: (angular) velocity.

if $a(t) \geq \alpha$, and $f(a(t)) = 0$ otherwise. Here $\alpha$ and $\beta$ are parameters of the activation function. The activation level vector $\boldsymbol{a}(t)$ (or $\boldsymbol{a}$) of the output layer is given by standard neural rate decay dynamics,

$$\tau \frac{d\boldsymbol{a}(t)}{dt} = -\boldsymbol{a}(t) + \sum_{j=1}^{n_I} g_j \boldsymbol{W}^{(j)}(t) \boldsymbol{f}^{(j)}(t), \tag{2}$$

where $\tau$ is the membrane potential time constant controlling the speed of natural decay of neural activation, $g_j$ is a gain constant controlling the amplitude of input from the $j$th input layer, $n_I$ is the total number of input layers (or brain areas), $\boldsymbol{f}^{(j)}$ is the firing-rate vector of the $j$th input layer, and $\boldsymbol{W}^{(j)}$ is the weight matrix of the connection from the $j$th input layer to the output layer. All elements in $\boldsymbol{W}^{(j)}$ are assumed to be non-negative (except the inhibitory self-connection in the HD attractor) [36]. Therefore, $g_j > 0$ indicates excitatory projections while $g_j < 0$ indicates inhibitory projections.

Non-plastic connections in the model include one-to-one feed-forward projections (e.g. between the attractor and gRSC) and imply the same number of neurons $N$ both in the input and output layer. Inhibitory connections can be of two types: global inhibition to all neurons within a layer

$$\boldsymbol{I}_{\text{GI}}(N) = \frac{\boldsymbol{J}(N)}{\sqrt{N}}, \tag{3}$$

and lateral inhibition originating from one neuron and equivalently affecting all other neurons in the same layer (e.g. as in Fig 1D, bottom, among aLB cells), i.e.

$$\boldsymbol{I}_{\text{LI}}(N) = \frac{\boldsymbol{J}(N) - \boldsymbol{I}(N)}{\sqrt{N-1}}, \tag{4}$$

where $\boldsymbol{I}(N)$ is an $N \times N$ identity matrix and $\boldsymbol{J}(N)$ is an $N \times N$ all-ones matrix. These connections are kept stable at all times, i.e. they are not subject to learning. See S1 Appendix for details and S2 Table for a summary of all neural connections used in this paper.

## The HD attractor ring

For simplicity we employ a single HD ring attractor (see S2 Appendix) [36], updated by angular velocity and consisting of $N_{\text{HD}} = 360$ HD neurons with equally spaced preferred HDs. Symmetric connections within the attractor (not subject to learning) sustain a stable Gaussian firing pattern without external inputs. During head turning (Fig 1C), asymmetric self-connections within the attractor, scaled by the magnitude of angular velocity, update the firing pattern of HD cells. That is, path integration is performed.

## The subcortical pathway

The HD attractor provides unimodal HD signals to gRSC, which are in turn projected further to dRSC for the integration with aLB signals (purple pathway in Fig 1A). Similar to the HD attractor, both gRSC and dRSC consist of neurons with different preferred directions, inherited from the upstream HD attractor, and each contains the same number neurons as the HD attractor.

For the connections among these structures (see S2 Table for a summary), the HD attractor and gRSC are connected by 1-to-1 connections, containing the same classic (unimodal) HD cells. gRSC and dRSC are fully connected (with initially random connectivity), subject to Hebbian plasticity (S1 Appendix; also in Fig 1A, marked as 'Hebbian'). This allows the global HD signal to be associated with environment-specific aLB signals conveying visual information for

HD calibration. Without plasticity in this connection, dRSC could only receive unimodal HD signals from gRSC, contradicting the existence of bimodal firing patterns (BD cells) observed in dRSC during darkness (when sensory inputs and hence aLB signals would be absent) after the rat experienced two environments with conflicting visual cues [28]. We also assume global self-inhibitory connections (Eq 3) among dRSC and gRSC in order to restrict the retrosplenial activation from spilling over.

## Feature-specific visual signals

Visual inputs from a specific scenery are separately provided through different feature-based channels (see Fig 1B for an illustration). That is, each channel selectively codes visual information based on its own preferred feature. The term 'feature' stands for perceptual attributes (e.g. colors, edges, contrast, luminance, etc.), possibly mediated via known anatomical projections from visual areas 17 and 18 to RSC [39,40]. Each feature-specific channel contains neurons with their own egocentric tuning directions $\theta$. The egocentric visual space of the agent ranges from -180° to 180° ('ahead' being 0°; Fig 1B), and changes with the direction of the agent.

We compose our sceneries of three basic types of visual signals. The first type is a scaled von Mises distribution with a single peak (a circular Gaussian):

$$\boldsymbol{f}_{\mathrm{vI}}(\varphi, \kappa) = \frac{f_{\max}}{e^{\kappa}} e^{\kappa \cos(\theta - \varphi)},$$

(5)

where $\kappa$ is the encoding precision and $\varphi$ is the preferred direction with maximum firing rate $f_{\max}$. The second type (e.g. the 'blue' cue in Fig 1B) extends $\boldsymbol{f}_{\mathrm{vI}}$ by enlarging the range of high firing rate, yielding a unimodal signal with less directional specificity, calculated as the cumulative distribution function of a von Mises distribution (i.e. summing multiple offset circular Gaussians with encoding precision $\kappa_1$ and a range of parameters $\varphi$). The third type (e.g. 'red' cues in Fig 1B with encoding precision $\kappa_1$) consists of two von Mises distributions with different peaks, yielding a multimodal firing pattern (e.g. the 'red' cue in Fig 1B). See S3 Appendix for details and Fig 2A for exact firing curves.

A key premise of the model is that visual signals for every feature (plus low amplitude background noise), if not zero for all directions, are scaled to the same mean, yielding fluctuation of firing rates about the overall mean stimulus intensity. This property is inspired by neural adaptation in sensory systems [41], hypothesized to occur upstream of the brain areas that comprise the model, and allows for a distinction of salient vs non-salient cues and associated novelty detection (see Results). To avoid parallax effects [27] and focus on properties of cue rich environments, all cues are treated as distal cues (i.e. egocentric bearing and HD are separated only by a constant, arbitrary offset).

## Abstract landmark bearing and the mOSA learning algorithm

We propose that abstract landmark bearing (aLB) cells sparsely encode the conjunction (i.e. the co-occurrence in space and time, or a 'local view') of all feature-specific visual inputs, driven by afferent projections from early visual areas. That is, a single aLB cell encodes the egocentric conjunction of all cues at a given orientation rather than encoding single cues separately. A population of aLB cells that is active in a given environment encodes all possible local views with a finite angular resolution, given by the total number of aLB cells active in that environment.

The firing of aLB cells is determined by their inputs from visual cells, via connections whose weights are adjusted using a version of Oja's subspace algorithm (OSA) [34,35]. The original OSA can extract principal subspaces as components from time-series information within a neural network architecture, compared to standard Hebbian learning an additional

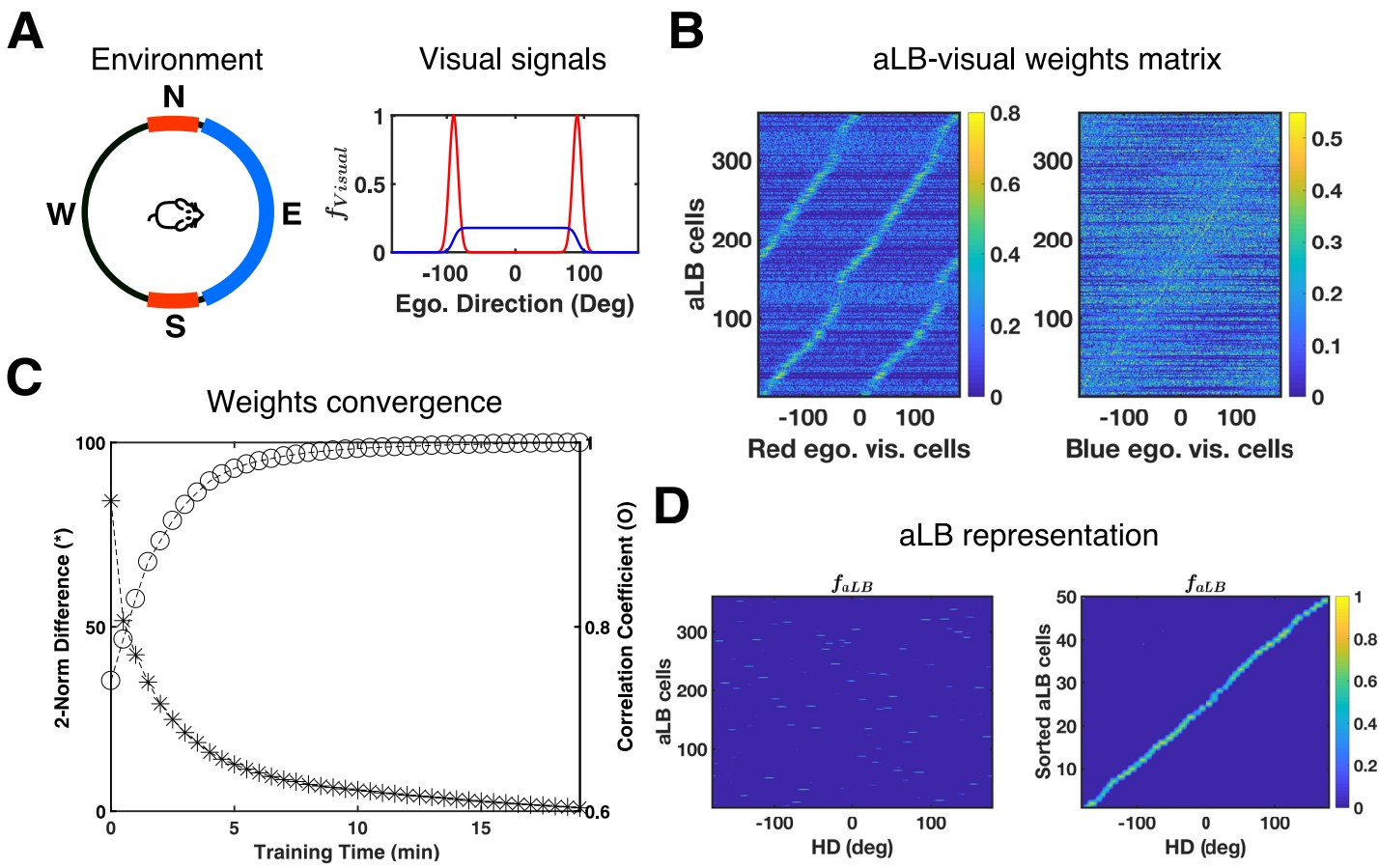

**Fig 2. aLB cells express a unimodal representation of landmark bearing via modified OSA.** (A) The landmark configuration and feature-specific V1 input signals (same as Fig 1B) for this simulation. Each curve corresponds to the visual cue with the same 'color'. (B) Sorted weights to aLB cells from visual cortical cells responding to the 'red' feature (left) or the 'blue' feature (right). The left figure indicates that aLB cells learn a bimodal profile with 'red' visual cells due to the conflicting 'red' cue. Warmer colors represent stronger synaptic connections. V1 cells are labelled by their own preferred egocentric directions. aLB cells are sorted according to which V1 cells they receive the maximum weight from. (C) The convergence of synaptic weights during learning. The asterisk stands for the two-norm (Euclidean distance) of the difference, while the circular marks stand for the Pearson correlation coefficient, between the vectors of current and final weight matrices. (D) The global representation of aLB cells (labelled by positive numbers on the y-axis) shows sparsity and unimodality during the testing phase (left). Sorting aLB cells according to the HD with maximum firing rate correlation (right) shows that aLB cells have a unimodal firing profile and cover the entire range of egocentric bearings, despite not following any one visual cue alone. Warmer colors represent higher firing rates. Abbreviations: HD: head direction; aLB: abstract landmark bearing; ego.: egocentric; vis.: visual; $f$: firing rate.

feedback term (for weight decrements) is present [34,42] (see below for details). Here the OSA acts between visual cells and aLB cells. Compared to the original OSA we introduce two modifications: 1. Lateral inhibition among aLB cells (Eq 6); and 2. Enforcement of non-negative weights from visual cells (Eq 7). We first describe each modification and then outline the rationale behind each and how the interplay of those changes yields the desired properties of aLB cells.

1. Lateral inhibition: aLB cells are subject to lateral inhibition among each other, allowing for winner-takes-all dynamics [43,44], similar to sensory pattern discrimination [45,46]. The decay dynamics of aLB cells can be written as

$$\tau_{aLB} \frac{d\boldsymbol{a}_{aLB}}{dt} = -\boldsymbol{a}_{aLB} + g_{aLB} \boldsymbol{W}_{aLB} \boldsymbol{f}_{aLB} + \sum_{j=1}^{N_F} g_{V2aLB}^{(j)} \boldsymbol{W}_{V2aLB}^{(j)} \boldsymbol{f}_{Visual}^{(j)}. \tag{6}$$

Here, $W_{\text{aLB}}$ is the matrix of constant lateral inhibition weights among aLB cells (see Eq 4), the subscript 'V2aLB' refers to the parameters for the connection from visual areas to aLB cells, and 'Visual' refers to the parameters for the related visual layer. For instance, $g_{\text{aLB}}$ is the gain factor controlling the amplitude of lateral inhibition weights among aLB cells, $W_{\text{V2aLB}}^{(j)}$ is the connection weight matrix between aLB cells and visual cells within the $j$th visual feature, and $f_{\text{Visual}}^{(j)}$ is the firing-rate vector of visual cells for the $j$th visual feature channel (red, green, blue in Fig 1). $N_{\text{F}}$ is the total number of visual features. All parameter values are summarized in S1 Table.

The original OSA does not assume lateral inhibition among postsynaptic neurons [34]. The main effect of lateral inhibition is to enforce winner-take-all dynamics among aLB cells and a yields sparse encoding of a given egocentric view of landmarks that is consistent with a small range of HDs (cf. Fig 1D). The sparse recruitment of different sets of aLB cells in different environments ensures high capacity (see Results).

2. Non-negative weights in the Oja's Subspace Algorithm: The synaptic weights $W_{\text{V2aLB}}^{(j)}$ in Eq 6 are updated via the modified version of Oja's Subspace Algorithm (mOSA) shown in Eq 7. The original OSA ensures global convergence without explicit weight normalization [34,35,42]. However, the original algorithm does not stipulate the sign of the synaptic weights. To ensure positive feedforward connectivity, $[\cdot]_+$ maps negative elements of the weight matrix to zero, since biological synapses do not change sign.

$$W_{\text{V2aLB}}^{(j)} \rightarrow [W_{\text{V2aLB}}^{(j)} + \eta_{\text{V2aLB}} f_{\text{aLB}} (f_{\text{Visual}}^{(j)} - W_{\text{V2aLB}}^{(j)}{}^T f_{\text{aLB}})^T]_+. \tag{7}$$

Here $\eta_{\text{V2aLB}}$ refers to the learning rate for all visual features. Note that the convergence of weights during learning is not affected by the $[\cdot]_+$ operator (see [47] for a theoretical interpretation of non-negative OSA).

The feedback term $W_{\text{V2aLB}}^{(j)}{}^T f_{\text{aLB}}$ in Eq 7 contains the transpose of the incoming weights multiplied by the aLB firing rates, corresponding to the expected feature-specific visual pattern that should yield the current aLB firing pattern. That is, it drives the plastic connections to be updated in ratio with the difference between real and expected visual inputs (i.e. $f_{\text{Visual}}^{(j)} - W_{\text{V2aLB}}^{(j)}{}^T f_{\text{aLB}}$). $\eta_{\text{V2aLB}}$ controls the learning rate of both the feedback and feedforward term and effectively introduces a threshold for potentiation, further enforcing sparseness (in addition to the initial effect of lateral inhibition). The feedback term also enables the disconnection (depression of weights) of activated aLB cells from inactive visual cells (when an expected feature is not present, i.e. when $f_{\text{Visual}}^{(j)} = 0$ in Eq 7). This renders the aLB population highly flexible, allowing it to update its encoding of the visual scene when changes to the visual world are introduced. Moreover, the feedback term supports the desired effect of the lateral inhibition, as it allows the winner-take-all dynamics among aLB cells to unfold before multiple aLB cells develop strong weights for the same local view. This is particularly useful if an ambiguous cue (such as the red feature in Fig 1D) provides residual input to an aLB cell that already codes for a different local view.

In summary, lateral inhibition and the feedback term in mOSA support sparseness and responsiveness to changes in visual inputs in the aLB population, while the restriction to non-negative weights respects the fact that synapses should not change sign. Note that mOSA (like the original OSA) is a non-local algorithm, but could be equivalently realized by a local algorithm in terms of extra interneurons encoding feedback signals (e.g. [48]).

Note that Eq 7 can be rewritten in the following form:

$$\boldsymbol{W}_{\text{V2aLB}}^{(j)} \rightarrow \left[\boldsymbol{W}_{\text{V2aLB}}^{(j)} + \eta_{\text{V2aLB}} \boldsymbol{f}_{\text{aLB}} (\boldsymbol{f}_{\text{Visual}}^{(j)} - \zeta \ \boldsymbol{W}_{\text{V2aLB}}^{(j)}{}^T \boldsymbol{f}_{\text{aLB}})^T\right]_+. \tag{7B}$$

Introducing the prefactor $\zeta$ ($>1$) in front of the negative feedback term can be compensated for by scaling down the learning rate $\eta_{\text{V2aLB}}$ and scaling up the gain factor $g_{\text{V2aLB}}$ in Eq 6 to yield the same outcome for all simulations reported in the Results section. However, the interpretation (and predictions) for what happens at the level of biological implementation changes. The prefactor $\zeta$ suggests that the weight decay is modulated by a feedback connection from aLB cells to visual areas. Conversely, down-regulating the scaling factor $g_{\text{V2aLB}}$ (compared to Eq 7B) potentially corresponds to a neuro-modulatory effect on the connection converging onto aLB cells from visual areas.

It should be stressed that other learning algorithms fail to yield appropriate feedback connection for HD in the complex environments considered here (including ambiguous cues). That is, the use of complex environments with multiple cues (some of which are multimodal or ephemeral, e.g. the red cue in Fig 1B) allows for model properties that are not evident in simpler setups. In other words, the failure of other learning algorithms to produce feedback signals appropriate to reset HD only becomes apparent by considering environments with cue configurations more akin to real-world environments (e.g. containing repeating or highly similar visual patterns). We compared mOSA to results obtained with other neural learning algorithms acting between visual and aLB layers. Neither Hebbian learning with lateral inhibition among aLB cells, nor Hebbian covariance learning [49,50], Intrator's BCM algorithm [51], or the original Oja Subspace Algorithm (i.e. OSA with no self-inhibition) [34] could generate unimodal aLB representations (and thus appropriate HD feedback) in the environment depicted in Fig 1B (see S4 Appendix for details and S1 Fig for simulation results of these alternative algorithms).

## The cortical pathway and dRSC

Receiving egocentric visual input from visual layers, aLB cells encode conjunctions of egocentric bearings on multiple features/cues. This signal is integrated with HD in dRSC, anchoring the global HD signals and egocentric visual signals to dRSC via Hebbian learning between aLB cells and dRSC as well as between gRSC and dRSC (see S2 Table for a summary of all neural connections used here), i.e.

$$\boldsymbol{W}(t + \Delta t) = \boldsymbol{W}(t) + \eta \boldsymbol{f}_{\text{dRSC}}(t) \boldsymbol{f}_{\text{Input}}(t)^T, \tag{8}$$

where the term 'Input' refers to either aLB or gRSC, with $\boldsymbol{W}$ and $\eta$ refers to the corresponding connection matrix and learning rate. During the test phase, we set $\eta = 0$ to fix synaptic connections. Weights subject to Hebbian learning are normalised to ensure their stability with this learning rule. More specifically,

$$w_l(t + \Delta t) \rightarrow \max\left\{\frac{w_{\text{max}}}{\|w_l(t + \Delta t)\|_2}, 1\right\} \cdot w_l(t + \Delta t) \quad \forall l, \tag{9}$$

where $\|\cdot\|_2$ is the two-norm for calculating the total connection, $w_{\text{max}}$ is the maximum total connection strength, and $w_l$ is the $l$th row vector (i.e., synaptic connections targeting the $l$th output neuron) of the weight matrix $\boldsymbol{W}$.

To stabilize the subcortical HD attractor against drift, a projection with both 1-to-1 excitation and global inhibition from dRSC transmits the integrated signal back to the HD attractor (S2 Appendix). Choosing dRSC as the locus of integration is coherent with experimental

results, showing that landmark-dominant HD signals (i.e. BD cells) were only discovered in the dRSC, instead of in the subcortical pathway (i.e. not in gRSC, anterodorsal thalamic nucleus, or dorsal presubiculum) [28].

### Visualization of HD and aLB representations

The representation of internal HD signals is naturally defined as the firing-rate weighted mean of preferred directions of HD neurons, i.e. the population vector, both in the attractor and gRSC. More specifically,

$$\bar{\theta}_{HD} = \arctan\left(\frac{\boldsymbol{f}_{HD} \cdot \sin\boldsymbol{\theta}}{\boldsymbol{f}_{HD} \cdot \cos\boldsymbol{\theta}}\right) \in [-180, 180), \tag{10}$$

as $\boldsymbol{f}_{HD}$ is the firing rate vector of the HD attractor, $\boldsymbol{\theta} = \{\theta_l = -180+(l-1)\Delta\theta | 1 \leq l \leq N_{HD}\}$ is the set of preferred directions, and $\cdot$ is the dot product. gRSC cells share the same representation.

To visualize the global representation of activated aLB cells, their firing rates during the testing phase are binned according to the real HD where they show maximum firing rates above a threshold $\varepsilon_{aLB} = 0.5$. Binning according to true HD reveals unimodality or multimodality of aLB firing patterns, even though aLB cells, unlike HD cell, are not presumed to have a certain preferred direction in advance.

We use the Jaccard similarity coefficient [52], or 'Intersection over Union' (IoU), to measure the similarity of neural firing patterns (over aLB cells) between different testing environments. The IoU is defined as the proportion of cells that is highly active in both environments (the overlap between two set of active cells) relative to the number of cells that is highly active in only one of the two environments being compared (i.e., the union of cells exclusive to one environments).

$$IoU = \frac{|\boldsymbol{S}_1 \bigcap \boldsymbol{S}_2|}{|\boldsymbol{S}_1 \bigcup \boldsymbol{S}_2|}, \tag{11}$$

where $\boldsymbol{S}_i = \{l | f_{aLBl} \geq \varepsilon_{aLB}, 1 \leq l \leq N_{aLB}\}$ is the set of activated aLB cells for the $i$th tested environment and $f_{aLBl}$ is the firing rate of the $l$th aLB cell.

### Simulations

Every simulation consists of the learning phase and testing phase. Learning with neural plasticity turned on is based on a 20-minute HD trajectory from rat recordings, with an average angular velocity of 72.88 deg/s (Fig 1C and S1 Dataset). In the testing phase, all synaptic changes are frozen and a slow uniform rotation with 60 deg/s for 60 seconds is simulated, starting from the final HD of the learning phase. This way each HD is sampled 10 times and a balanced estimate of the learned representation across all directions is sampled for visualization. All simulations are conducted via MATLAB R2017a.

### Results

We first examine the capability of aLB cells to form a landmark representation from unstable cues within an environment and the capacity of aLB cells to learn distinct sets of landmark representations across multiple environments. We then show how visual/sensory feedback may be conveyed from aLB cells to HD cells via RSC, and how the model is coherent with recent empirical findings. Finally, we consider model-specific predictions.

## aLB cells with unimodal encoding in complex sceneries

We first show how aLB cells learn a sparse and unimodal representation over a complex scenery. Visual projections to RSC arise in early visual areas [26], and we do not posit upstream curation of landmark inputs. Instead, the learning algorithm must generate an egocentric sensory representation suitable to convey feedback to HD cells. We set up two visual landmarks in a single environment (Fig 2A, same as Fig 1B): an unambiguous, broad 'blue' cue located due East with low encoding precision, and conflicting, narrow 'red' cues located due North and due South. The simulation is based on the HD trajectory shown in Fig 1C. The synaptic weights (Fig 2B) between visual areas and aLB cells quickly converge (Fig 2C), and the unimodal representation of aLB cells stably emerges at an early stage of learning (S2 Fig). Moreover, nearly all aLB cells are active for only one egocentric bearing (Fig 2D, right). aLB also exhibit high encoding precision for each preferred direction (Fig 2D). This aspect is extracted from visual signals with high encoding precision and narrow firing range ('red' cues in Fig 2A), and shows that aLB cells integrate useful properties from multiple cues (i.e. high encoding precision from the two narrow cues and unidirectionality from the isolated wide cue). Overall, these results suggest that the algorithm successfully achieves a suitable unimodal aLB cell representation when exposed to scenery with multiple and ambiguous features.

## Robustness against unreliable information

aLB cells can maintain unimodal encoding of a complex scene in the presence of unreliable visual information without any extra top-down control (e.g. visual attention). Unreliable sensory inputs in our simulations include landmarks moving with a constant speed, cues that vanish and reappear somewhere else (i.e. teleported cues), or ephemeral cues that only temporarily exist.

First, for the case of moving cues, we set up the same single environment as Fig 2A but with the unambiguous 'blue' cue moving along the cylinder at an observable speed of 90 deg/s anticlockwise (Fig 3A, left). No unimodal aLB representation develops in this case when tested in the same environment with the moving 'blue' cue (Fig 3A, middle). However, freezing the developing weights and stabilizing the moving cue yields an approximately unimodal aLB representation (Fig 3A, right), suggesting that the learning algorithm tries to maintain a set of weights coherent with the most recently experienced scene containing both features.

Second, for the case of teleporting cues, we set up the same environment but with the unambiguous landmark periodically teleporting to a random position on the cylinder (every 10 seconds; Fig 3B, left). Similar to the case of moving cues, we find an approximately unimodal aLB representation once the developing weights are frozen and the teleporting landmark is stabilized, without additional learning once the landmark stops teleporting (Fig 3B, right). Together these results suggest that the mOSA algorithm can continuously update the weights and unlearn weights from unstable cues (locations in the field of view occupied prior to the most recent teleportation event) and extract weights from stable cues. Thus the model exhibits the preference for encoding stable cues, consistent with increased fMRI signals in RSC for more stable landmarks [53].

Note, the fact that the same set of active aLB cells is maintained throughout the above simulations depends on the properties of the visual inputs. Given the relatively low encoding precision of the 'blue' cue (and hence lower maximum firing rate given its width, see Materials and Methods) when the unstable 'blue' cue moves, the remaining stable 'red' cue can enforce the persistence of the same set of aLB cells through all cue-configurations without recruiting new aLB cells. This same set of cells simply develops incoming weights from visual cells coding for the updated position of the blue cue, while unlearning (see the learning rule of mOSA, Eq 7)

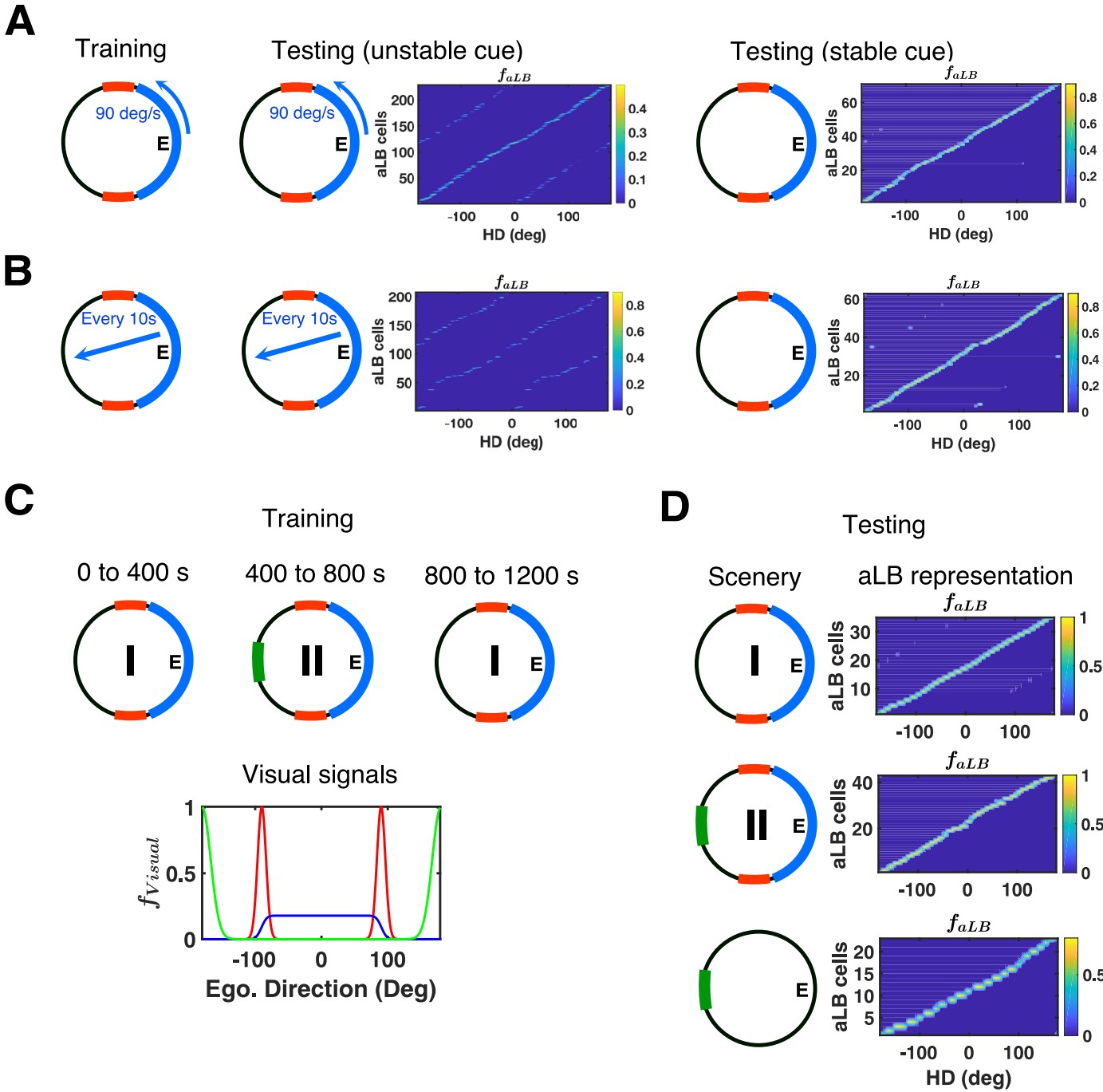

**Fig 3. aLB cells exhibit robustness against unstable or ephemeral cues, and can incorporate novel cues.** (A) Single environment (same as Fig 1B) with the 'blue' cue moving along the cylinder with 90 deg/s anticlockwise during training (left). Global representation of aLB cells tested on the unstable scenery with 'blue' cue moving by 90 deg/s anticlockwise (middle, $\varepsilon_{aLB} = 0$) and the stabilised scenery with 'blue' cue fixed after learning has stopped (right, $\varepsilon_{aLB} = 0.5$). (B) Single environment (same as Fig 1B) with the 'blue' cue teleporting to a random position on the cylinder every 10 seconds (left). Global representation of aLB cells tested on the unstable scenery with 'blue' cue teleporting every 10 seconds (middle, $\varepsilon_{aLB} = 0$) and the stabilised scenery with 'blue' cue fixed after learning has stopped (right, $\varepsilon_{aLB} = 0.5$). (C) Two sceneries (within the same environment) with the 'red-blue' scenery (Sc. I) and a 'red-blue-green' scenery (Sc. II). The agent stays in each scenery for 400 seconds (once from Sc. I to Sc. II and back to Sc. I), with feature-specific visual input signals in Sc. II (bottom). (D) Following the learning phase, the model is tested on the 'red-blue' scenery (i.e. Sc. I, top), the 'red-blue-green' scenery (i.e. Sc. II, middle), and the 'green' scenery (i.e. Sc. II with 'red-blue' scenery excluded, bottom). Stable unimodal aLB representations emerge in all cases (also see S3 Fig), with different sets of active cells in Sc. I vs. Sc. II (S4 Fig). Warmer colors represent higher firing rates of aLB cells. Abbreviations: HD: head direction; aLB: abstract landmark bearing; ego.: egocentric.

weights from visual cells coding for the previous position of the blue cue. That is, when the blue cue moves, the drive via the initial random connections from the 'blue' visual cells covering the new position is insufficient to override the currently active set of aLB cells receiving input from the stable 'red' visual neurons. Naturally, adding more stable cues will make the current set of active aLB cells even more resistant to change (as might be expected in cue-rich natural environments). On the other hand, adding a novel cue with saliency comparable or higher than existing cues will lead to the recruitment of a new set of aLB cells (see the next subsection).

### Incorporating novel cues

Here we consider the edge case where a newly added cue is of similar salience as the previously present array of cues. We set up an environment with the stable, conflicting 'red' cues and the unambiguous, broad 'blue' cue, similar to previous simulations. However, a novel and unambiguous 'green' cue (with encoding precision $\kappa_2$) appears due West after 400 seconds for a similar duration (Fig 3C). In this way, the agent is first confronted with the 'red-blue' scenery for 400 seconds, then the 'red-blue-green' scenery for another 400 seconds, and then again with the 'red-blue' scenery for the next 400 seconds.

After 1200 seconds, we tested the model on both the 'red-blue' scenery (Fig 3D, top) and the 'red-blue-green' scenery (Fig 3D, middle). Unimodal representations of abstract landmark bearing are successfully expressed in both cases, however with different sets of active aLB cells (with little to no overlap, S4 Fig). Contrary to the previous section, the inputs from the 'red-blue' scenery cannot enforce the persistence of the same aLB cells, because once the high-saliency green cue appears, it creates a strong drive to random aLB cells via its initial random connections, silencing other aLB cells via lateral inhibition in the aLB layer. This leads to the recruitment of a new set of aLB cells (S3 and S4 Figs). This may be viewed as the model treating the new 'red-blue-green' scenery as a novel environment, and this novelty detection is dependent on the cues being normalized to equal total firing.

Moreover, aLB cells also express a unimodal representation when tested on the 'green' cue alone (with the 'red' and 'blue' cues removed, Fig 3D, bottom), which has not been explicitly learned on its own before. This shows that the aLB representation is robust to cue removal on one hand, and that the model can incorporate a novel visual feature (even in the presence of already learned cues) into its landmark bearing representation on the other. Hence blocking by previously learned features is not an issue for the model, which is similar to place cells in the hippocampus [54].

Also note, the ability to incorporate a newly added cue is distinct from the question of whether or not a new set of aLB cells is recruited. If the newly added cue is of low saliency (i.e. of low directional specificity, with low peak firing rates among feature-specific visual cells), no new aLB cells are recruited (compared to the previously active ensemble) but the cue nonetheless contributes to driving previously active aLB cells following continued learning.

### High capacity of scene-based encoding

We suggest that the present model approximates the learning of HD feedback signals from sensory inputs in environments that are richer than stereotypical simulation environments with a single Gaussian cue, and hence more indicative of the mechanisms at play in rich real-world scenes. If this is the case, the model should be able to cope with a large number of environments concurrently. That is, the learning rule should ensure a large capacity for distinct environments, using a limited number of neurons overall, and recruit new sets of aLB cells in

new environments to provide unique feedback connections to HD cells without interference from or forgetting of previously learned feedback for a different environment.

To test the capacity of the model, we set up 10 environments that the agent experiences sequentially (Fig 4A). All environments share the same stable unambiguous narrow 'red' cue and stable unambiguous broad 'blue' cue. The distinguishing cue between all 10 environments is an additional unambiguous narrow 'green' cue, occupying a different position in each environment (starting due West in the first environment) and shifted by 36˚ anticlockwise in each successive environment. During the learning phase, the agent stays in each environment for 120 seconds before moving to the next environment. In these simulations continuity between the environments is assumed. That is, the HD at the last moment in environment 1 is the starting HD in environment 2, as if the agent went from one environment into an adjacent one.

To quantify the similarity (or lack thereof) among recruited sets of aLB cells across different environments, we test the model (in each environment) with snapshots of the learned synaptic connections between visual cells and aLB cells. Testing the model generates a set of active aLB cells for each synaptic snapshot (Figs 4B and S5). Similarity between the active sets of aLB cells across environments is quantified via IoU similarity maps [52].

The first comparison is based on snapshots of synaptic weights taken after the first exposure to each environment ('intermediate vs. intermediate' in Fig 4C), i.e. we quantify the similarity between the recruited aLB population with one set of weights (e.g. immediately after learning environment I) and the recruited population of aLB cells with another set of weights (e.g. immediately after learning environment II).

The second comparison is based on the sets of recruited aLB cells with synaptic weights taken after learning has ended ('final vs. final' in Fig 4C), i.e. once all 10 environments have been experienced. In both cases, the similarity between sets of recruited aLB cells is maximal (i.e. 1) when tested for the same environment and drops to near-zero when tested for all distinct environments, even with the smallest separation of the green cue between 'adjacent' (in time/cue separation) environments, suggesting aLB cells providing distinguishable signals for multiple environments even when these environments share some cues.

To quantify the similarity among recruited sets of aLB cells for the same environment across time, the third comparison is based on the sets of recruited aLB cells with synaptic weights taken after the first exposure to each environment, vs. the sets of recruited aLB cells with synaptic weights taken after the entire learning period ('intermediate vs. final' in Fig 4C). For all 10 environments tested, we find the high similarity between 'intermediate' and 'final' sets of recruited aLB cells within a given environment, even when other, distinct environments, have been learned in the meantime. This suggests the set of aLB cells recruited upon first exposure is preserved over time. Apart from the capacity to maintain distinct sets of visual-to-aLB feedback connections Fig 4 also demonstrates the ability of the system to perform pattern separation, given the similarity of the environments. S9 Fig shows similar results for a more varied set of cues in each environment. The capacity also scales with the number of aLB cells (S9C Fig).

Together the above results suggest a high capacity to maintain distinct feedback connections for a large number of environments. Since aLB cells convey visual feedback to HD cells (via RSC; see next sections), these distinct feedback connections (from distinct sets of aLB cells) can ensure HD remapping between distant environments and continuity between adjacent environments [55]. The effective remapping (in 1D) of aLB cells is reminiscent of place cell remapping (in 2D) [56,57]. That is, when an animal is re-introduced into a familiar environment, the sparse subset of aLB cells specific to that environment will be recruited to convey feedback, similar to the associative memory recall shown in hippocampus [58], and HD cells will adopt the preferred firing direction they exhibited previously in any given environment.

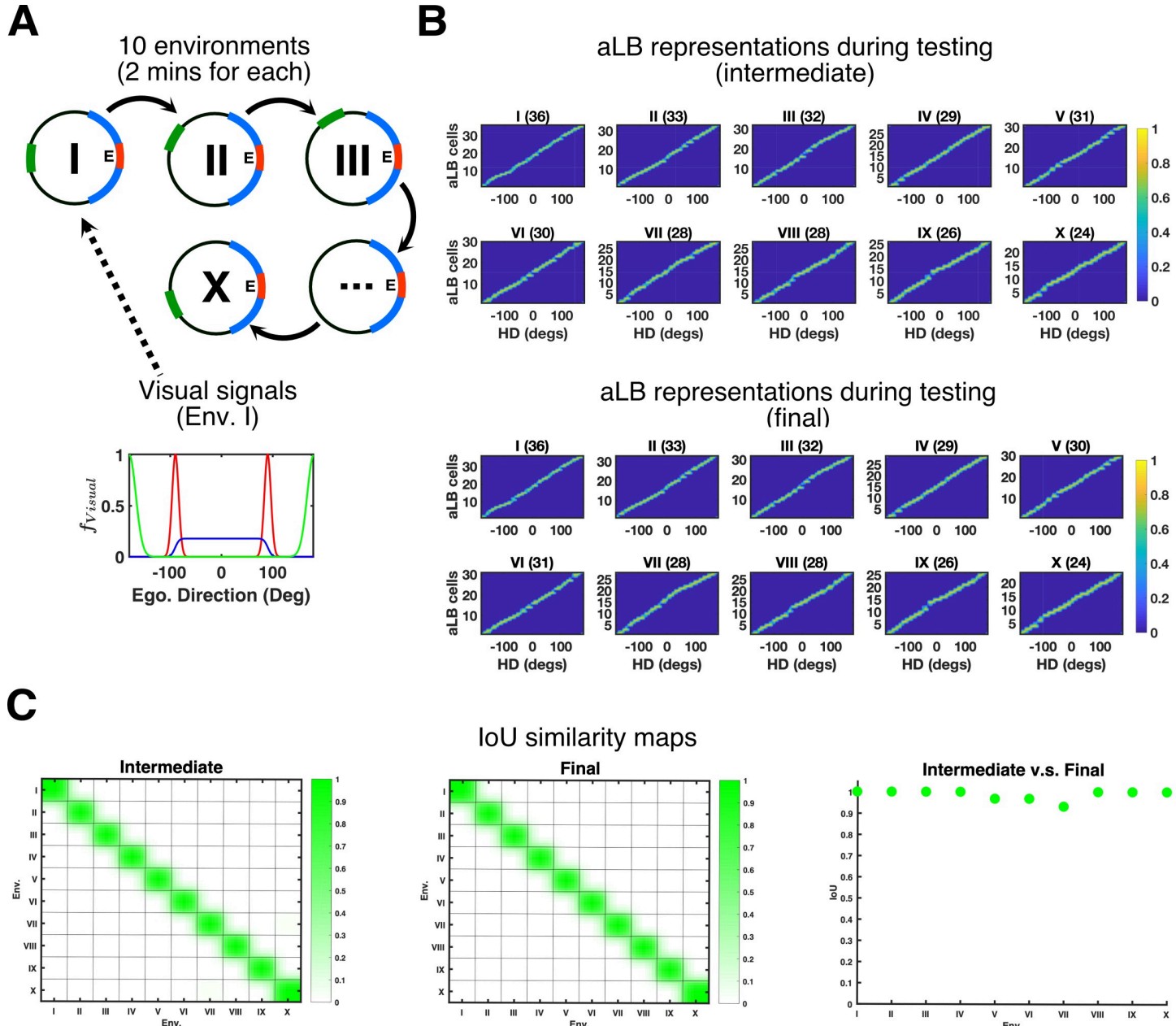

**Fig 4. aLB cells show high encoding capacity across multiple environments.** (A) The agent is exposed to 10 environments sequentially (top) with feature-specific visual input signals due East in Env. I (bottom). Sceneries contain 'red' and 'blue' cues, common to all environments, and a 'green' cue, at an orientation relative to the other two cues that change between environments. (B) Global representations of aLB cells with local weights tested on corresponding sceneries. Titles for each plot refer to the corresponding environment in (A), along with the total number of highly activated aLB cells in brackets. The 'intermediate' plots provide aLB cell activity based on weights after the first exposure in each individual environment. The 'final' plots provide aLB cell activity based on weights after the learning in all 10 environments is complete. See Fig 2D for illustrations. Warmer colors represent higher firing rates. (C) IoU similarity maps measuring the similarity of two sets of aLB cell firing patterns when tested on specific sceneries. Axes refer to environments, with 1 as the earliest (i.e. Env. I). The left column provides IoU maps based on 'intermediate' weights. The middle column provides IoU map based on 'final' weights. The right column provides the IoU index (y-axis) between the first exposure and the end of learning tested on each scenery (x-axis). Green colors represent higher IoU, of which the maximum is 1. Abbreviations: HD: head direction; Env.: Environment; Ego.: Egocentric; aLB: abstract landmark bearing; IoU: Intersection over Union; *f*: firing rate.

## Multimodal dRSC signals across mirrored environments

To investigate how the processed landmark information from aLB cells can act on HD cells, and to check for consistency with experimental data [28], the sensory aLB cells are now connected with dRSC, which receives HD information from the HD attractor (via gRSC) and in turn sends feedback to the attractor to compensate for natural drift (see Fig 1A for the full network).

Constraints on the systems-level function of RSC in HD processing come from recent experiments reported by Jacob et al. [28]. In these experiments, rats could freely explore two environments with an identical simple visual cue card located in both two environments but in opposite directions (e.g. West and East, respectively) with distinct odor cues in each environment. After exploration across these two simple environments, two types of bidirectional (BD) HD cells were reported. Some cells fired for two HDs, one per environment. These cells were termed between-compartment BD cells (BC-BD). In addition, within-compartment BD cells (WC-BD) showed bimodal HD firing patterns when restricted in a single environment. Both BD cell types (BC and WC) were solely found in dRSC.

To test that the aLB model of landmark processing is compatible with these findings, we set up two connected environments similar to the experimental design from Jacob et al. [28] (also see [30] for related modelling work). Both environments were additionally enriched with multiple cues constituting complex sceneries (Figs 5A and S6A). The second environment has the same scenery but is rotated by 180°. In our simulations, the agent alternates between Env. I and Env. II on a short time scale (relative to the whole learning time) of 60 seconds (Fig 5A), mimicking the free access to both environments experienced by the animals in Jacob et al.'s

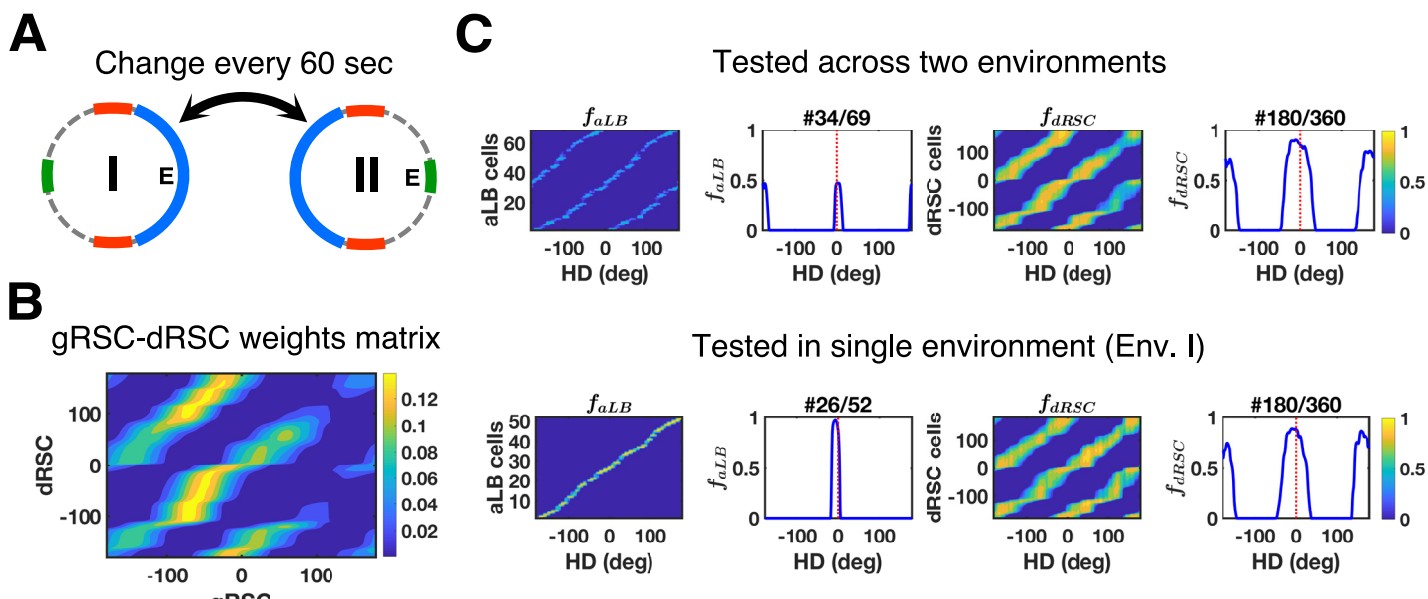

**Fig 5. Mirrored environments with the simulation result in dRSC signals.** (A) Illustration of the connected environments with mirrored and rotated complex sceneries. The agent repeatedly crosses from one environment to the other, spending 60 seconds in each environment at a time, for the duration of the whole 20-minute learning phase. (B) The learnt connection weights form the bimodality of dRSC responses after the whole learning phase. (C) Global representations of aLB cells (left, with a single cell example) and dRSC cells (right, with a single cell example) after learning, tested in alternating environments (top, row; 10 seconds in each environment) or in a single environment (Env. I, bottom row). Note that aLB cells are unimodal within a given environment (bottom left), contrary to WC-BD. aLB cells are labelled as positive numbers, whilst other cells are labelled from -180 to 179 according to their own initial preferred directions (y-axis). The x-axis stands for HD. Abbreviations: aLB: abstract landmark bearing; gRSC/dRSC: granular/dysgranular retrosplenial cortex; HD: head direction; $f$: firing rate.

experiment [28]. Similar to earlier simulations, the learning phase is followed by testing phases where no more learning occurs.

After learning, neurons in dRSC show bimodal firing patterns during the testing phase when the agent alternates between both environments (Fig 5C, top). Coherent with experimental findings on WC-BD cells, dRSC cells in the model show a bimodal activity profile firing for the duplicate (arrays of) visual landmarks in opposite directions. Moreover, bimodal firing patterns are preserved when the agent is restricted to one of the environments for testing (Figs 5C, bottom, and S6B), as the learned connections between gRSC and dRSC preserve bimodal firing patterns when restricted in a single environment (Fig 5B). I.e. with HD coming into gRSC, the agent is exposed to two landmark signals across the two environments (one being the 180˚ rotated version of the other), leading to the same HD being associated with two landmark bearings. In addition, similar multimodal firing profiles of dRSC cells are also found in the simulation with three conflicting environments (S5 Appendix and S7 Fig), replicating a similar simulation result from [30]. These results suggest that the simulated dRSC cells in our model are consistent with the WC-BD cells reported by Jacob et al. [28], integrating abstract visual information with internal HD signals.

Jacob et al. [28] also used distinct odor cues in the two connected environments. To simulate olfactory cues, 'odors' were treated as constant 'visual' signals (spanning 360˚), distinct from the red, blue, and green input channels in Fig 5. Each environment is accompanied by a unimodal 'green' visual cue in opposite directions (S6C Fig). After alternating exploration, dRSC cells still show bimodal firing patterns when tested in any single environment (S6D Fig), i.e. bimodal cells are still of the WC-BD type. This suggests that BC-BD cells are unlikely to be solely explained by different odor cues, as modelled here. Neither are they likely to stand for aLB cells themselves with observable unimodal firing patterns, since BC-BD cells preserve bimodal firing patterns during darkness and are thus not solely of sensory origin [28], unlike aLB cells, which are driven by visual signals alone in the present model. The bimodal (WC-BD) firing patterns of many dRSC cells are also preserved when tested in darkness given unimodal odor cues in each environment (S6E Fig), in accordance with such experimental findings.

To test whether BC-BD cells could be generated by other means, we explore heterogeneous learning rates. For half of dRSC cells (180 out of 360), the learning rate on incoming connections is reduced to near zero (see S1 Table for value settings). Under these conditions, the model generates both WC-BD and BC-BD cells simultaneously (S6F Fig). This suggests that experimentally reported BC-BD cells [28] may correspond to dRSC cells that have not (yet) fully learnt bimodal firing in a single environment (S6F Fig, middle) due to variations in the learning rates across all dRSC cells (also see Discussion section).

## Stabilizing global HD signals

Compared to the present model, the majority of previous studies presumed a direct projection from visual cue representations to the HD attractor (e.g. [36]). Only recently the role of RSC has been considered in the stabilization of HD signals (e.g. [27,30]). Since we suggest a unimodal representation of complex sensory environments via aLB cells as opposed to direct projections from cue representations, we test whether aLB signals can indeed stabilize HD signals to correct accumulated path integration error.

In contrast to the natural drift without sensory inputs (Fig 6A), Fig 6B shows subcortical HD signals from the simulations in the previous section (cf. Fig 5). Across the whole learning phase, the HD representation as well as the HD firing field does not incur drift (Fig 6B, compared to red dashed lines indicating no change to the expected HD change). Feedback is

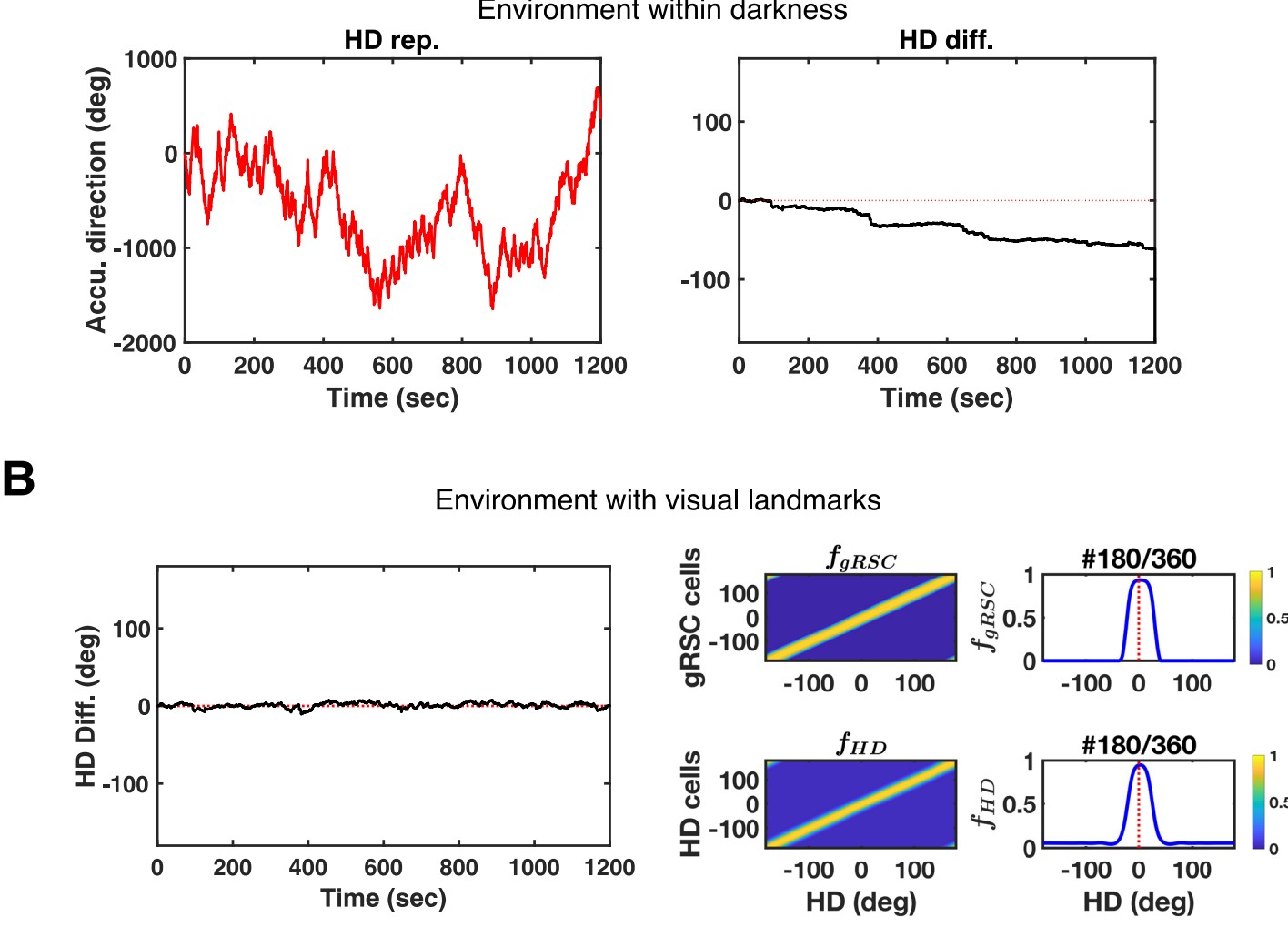

**Fig 6. dRSC signals stabilize global subcortical HD signals for the simulation in Fig 5.** (A) Simulation in darkness (left, no visual inputs resulting in no feedback from dRSC) shows systematic drift. The HD representation (left, red curves) and its angular difference from actual HD (right, black curves with actual HD subtracted from HD representations) are given over the whole 20-minute learning time. Red dashed lines indicate an ideal error-free HD representation. (B) Simulations of alternating exploration over the multi-environment case in Fig 5A showing stable HD representation (left), as well as global representations of gRSC and HD cells after alternating exploration showing no firing field drift (right). Red dashed lines stand for the center of initial HD firing fields. See the caption of Fig 5 for details. Abbreviations: HD: head-directional/head direction; Accu.: accumulated; Rep.: representation; Diff.: difference; gRSC: granular retrosplenial cortex; *f*: firing rate.

reliably conveyed by the learned connections, so that the accumulated path integration error in global HD signals is corrected via the feedback connection from dRSC to the attractor (pink connection in Fig 1A). That is, the visual information that is conveyed via aLB signals and integrated with HD signals in dRSC can correct the accumulated path integration error in subcortical HD cells.

Note that bimodal dRSC signals (i.e. WC-BD cells) do not disturb the unimodality of the subcortical HD representation. For example, if a WC-BD dRSC cell fires for North and South, while the simulated agent is facing North, the HD attractor would get North-specific feedback. The same bimodal dRSC cell fires when the HD is South. However, South-specific feedback from that dRSC cell would be compensated by corresponding feedback from another dRSC cell directed at the now active HD cells. Thus HD is not affected adversely.

## aLB cells vs. a direct visual-RSC connection

The core feature of aLB cells is their unimodal encoding of an array of multimodal sensory signals. These cells code for the conjunction of multiple cues for a given egocentric view. To test whether the subcortical HD system indeed benefits from such 'scenery-based encoding', we compare the model to an alternative model without aLB cells that employs a direct projection from visual areas to dRSC subject to classic Hebbian learning [27,30]. Crucially we employ a complex environment of the sort used above for both comparisons, since the benefits of aLB cells only become apparent in such environments.

The agent starts out exploring the same environment as in Fig 2A. After 10 minutes, the scenery is rotated by 120˚ anticlockwise (Fig 7A). If the array of landmarks is not treated as a coherent whole (i.e. a scene), the 'red' cue on its own can be perceived as having either rotated 60˚ clockwise, or 120˚ anticlockwise consistent with the 'blue' cue rotation. If such a conflicting 'red' cue has higher saliency, its contribution to feedback can dominate if a direct connection from visual areas to dRSC is employed in lieu of aLB cells (Fig 7C, top). For currently active HD cells receiving such conflicting feedback, preferred HDs 60˚ clockwise lie closer on the HD attractor ring than preferred HDs 120˚ anticlockwise. Thus, the HD signal is more likely to shift towards the clockwise direction (Fig 7C and 7D, right). That is, if individual cues in an array possess high saliency and ambiguous directional specificity, they may lead to this kind of erroneous feedback. With aLB signals, on the other hand, HD cells can correctly follow the 120˚ anticlockwise rotation (Fig 7B and 7D, left) as the whole scene (all cues) is encoded as a conjunction of cues.

Introducing complex cue arrays suggests Hebbian learning acting on a direct visual-to-dRSC connection cannot guarantee HD firing field rotation following environmental changes in complex environments. Together with the added computational power bestowed by aLB cells (robustness, the ability to disconnect themselves from unstable cues, resistance to blocking, and high capacity), these results suggest that a neural representation akin to aLB cells may serve as an intermediate processing layer for the scene-based sensory information within RSC or upstream of it.

To further test the utility of aLB cells, we compare the two models (one with aLB cells, one with a Hebbian visual to RSC connection) under the same multi-environmental case as in Fig 4 (capacity simulations). Each model is simulated in all environments sequentially, 2 minutes per environment. The distinguishing 'green' cue shifts by 36˚ anticlockwise in each successive environment.

Since distinct sets of aLB cells convey scene-based landmark bearing from multiple scenes to dRSC (with high capacity; Fig 4), different sets of aLB cells are recruited among these environments to provide unique feedback to dRSC without interference (Figs 4C and S5). For visualization purposes we test for environment-specific feedback by choosing HD between environments to be continuous. That is, the initial HD in each environment is different, whilst the HD at the last moment in Env. I is the same as the initial HD in Env. II. As a consequence, the preferred directions of HD cells (and gRSC cells) should not change across all environments if each environment provides the correct feedback signals. This is the case with aLB cells (Fig 8A–8C, left). That is, aLB cells have anchored new scenes to the current HD such that the correct HD is retrieved on each previously learned scene, with minimal heading errors (-2.66˚ ± 0.03˚).

However, this is not the case with a direct visual-to-dRSC Hebbian connection (cf. Fig 7), in which the preferred direction of dRSC cells (Fig 8A, right), as well as the HD cells (Fig 8B, right), would shift to follow the single changing cue (note that only 1 out of 3 cues shifts). In the absence of the conjunctive encoding provided by aLB cells, connections from the salient

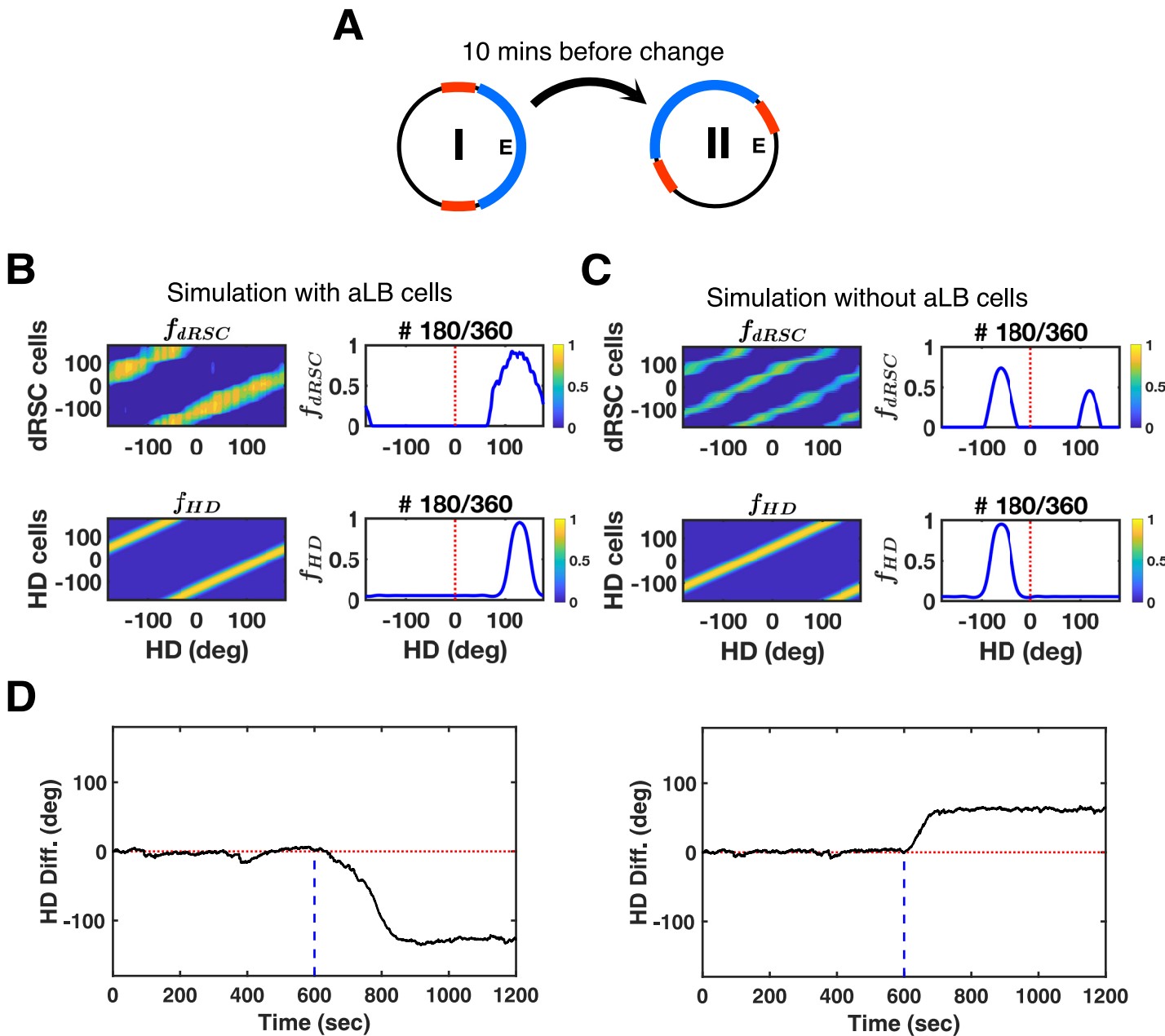

**Fig 7. Testing scenery-based HD encoding at a sensory level with the two-environment scheme.** (A) Sceneries in the first environment (Env. I) are all the same as Fig 2A, yet are rotated by 120° anticlockwise in the second environment (Env. II). Simulations are conducted within 20 minutes, assuming the agent explores each environment for 10 minutes. (B) Global representations of dRSC cells (top) and HD cells (bottom) tested on Env. II. Simulations are conducted via our two-stage model with aLB cells. (C) Global representations of dRSC cells (top) and HD cells (bottom) tested on Env. II. Simulations are conducted via the alternative model, shows an opposite direction of firing field drift from (B). This model does not contain aLB cells but rather a direct visual to dRSC connection (subject to classical Hebbian learning). (D) Simulations over the whole 20-minute learning time via our two-stage model with aLB cells (left) and the alternative model without aLB cells (right), showing opposite directions of HD drift. Abbreviations: HD: head direction; Diff.: difference; dRSC: dysgranular retrosplenial cortex; *f*: firing rate.

and changing 'green' cue learned in one environment will perturb HD in the next (cf. Fig 4). That is, HD cells would change their preferred directions between connected environments, despite the continuity of externally measured head direction (Fig 8B and 8C, right). This gives rise to large HD retrieval errors (151.91° ± 0.04°). Thus, we suggest that aLB cells (providing a

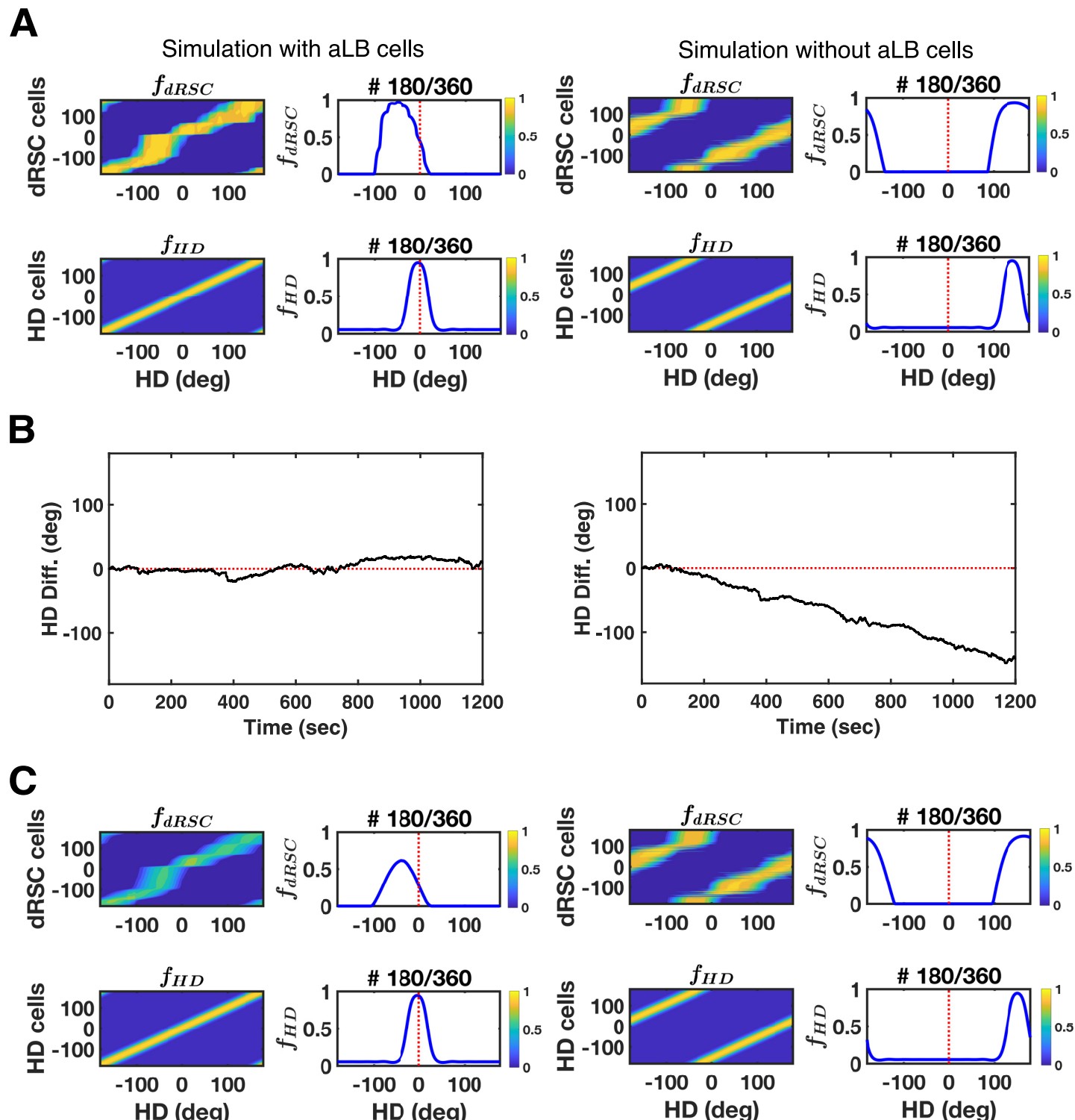

**Fig 8. The retrieval utility of aLB cells under multi-environmental exploration similar to Fig 4.** Simulations are conducted via our two-stage model (left, with aLB cells) and the alternative model (right, without aLB cells but rather a direct visual to dRSC Hebbian connection). (A) Global representations of dRSC cells (top) and HD cells (bottom) tested on the scenery in the final environment (Env. X). Red dashed lines stand for the center of initial firing fields. See the caption of Fig 4 for more details. (B) The difference between the actual HD and the HD representation over the whole 20-minute learning time. (C) Global representations of dRSC cells (top) and HD cells (bottom) tested on the sceneries in the first environment (Env. I), referring to new firing fields of dRSC and HD cells for retrieving the earliest scenery after learning across environments is complete. Abbreviations: Env.: Environment; dRSC: dysgranular retrosplenial cortex; HD: head direction; Diff.: difference; $f$: firing rate.

scene-based, conjunctive encoding) can enable HD retrieval with near zero HD errors across multiple sceneries that are partially overlapping at the sensory level, contrary to a direct V1-dRSC connection.

## Discussion

Head direction models have traditionally focused on the nature of the HD attractor circuitry (e.g. [36]). Visual feedback that anchored the attractor against drift is commonly represented as a simple Gaussian function, approximating classical HD recordings with a single cue card in the environment. Only recently, models have begun to investigate the mechanisms by which visual feedback for HD is conveyed in any detail [27]. Here we have proposed that abstract landmark-bearing (aLB) cells represent the conjunction of all available sensory cues. Produced by a modified version of the Oja's Subspace Algorithm [34], aLB cells can cope with more complex environments, with individual cues potentially being multimodal (ambiguous with regard to directional feedback), or of low directional specificity. We have shown that landmark processing by this algorithm can extract unimodality and directional specificity from distinct cues where neither cue by itself combines both directional precision and unimodality. The learning is robust with regards to unstable cues, can incorporate novel cues, endows the model with a high capacity for distinct environments, and is compatible with the most recent experimental data on retrosplenial HD coding.

### Abstract landmark bearing (aLB) cells

The strongest prediction of the model is the presence of a landmark signal that is purely sensory and unimodal, and that is not necessarily referencing a specific landmark, but rather the bearing of an array of cues, potentially including multimodal cues and cues with low directional specificity. This signal is encoded by cells we termed abstract landmark bearing (aLB) cells. Here, these cells encode scene specific bearings derived from visual signals, though it is conceivable that other sensory modalities contribute to their firing. aLB cells could contribute to panoramic memory supported by RSC [59]. Crucially, the firing of aLB cell constitutes a signal wholly dependent on sensory inputs and independent of vestibular inputs, unlike HD cells. This property could be tested in experiments that dissociate visual and vestibular signals, e.g. with sub-threshold vestibular rotations of an animal relative to its environment [60]. In addition, being of sensory origin aLB cells should cease firing in darkness, unless their firing is supported by sensory modalities other than vision. However, given the winner-take-all dynamics at play in the aLB layer, these cells may have a narrower tuning width compared to retrosplenial HD cells.

The present model is agnostic as to the exact location of these cells. Anatomically, dRSC holds strong reciprocal connections with visual areas [61] and parietal cortex [62], which are believed to code for the egocentric visual scenery prior to RSC [26]. In addition, besides RSC [10], HD-specific signals were also found in human medial parietal cortex [16], prestriate cortex [17], and deep layers of V1 [18] in rats. RSC has been implicated in sensory integration [63]. However, the classical subdivision of RSC into dysgranular and granular regions also hides a more fine-grained anatomical parcellation [64]. Therefore, aLB cells could be located in any of these cortical regions. It also remains a possibility that both direct and indirect pathways (with only the latter containing aLB cells) map visual inputs onto RSC concurrently.

### Lateral inhibition and feedback transmission

A clear role of aLB cells is that they represent a compressed representation of the entire visual scene, which is realized through lateral inhibition and the modified Oja's Subspace Algorithm

(mOSA). Neither the lateral inhibition alone (paired with classical Hebbian learning) nor the feedback alone (i.e. original OSA without lateral inhibition among aLB cells) yields a unimodal representation if the simulated agent is confronted with a complex scenery that contains conflicting cues. This suggests both lateral inhibition and a reasonably strong feedback term in the learning rule are necessary.

The feedback-term in mOSA implements long-term depression on the synaptic weights from individual visual cue representations (input channels) to aLB cells. This allows the algorithm to unlearn connections that may interfere with encoding, e.g. pruning initial random connections and disconnecting aLB cells from visual cells that represented an ephemeral cue. This suggests that pruning visual connections based on 'expected inputs' (conveyed by the feedback term) is necessary for the development of an aLB-like representation. Alternative algorithms, such as in Hebbian covariance learning [49,50] or Intrator's BCM [51] did not yield a unimodal representations of landmark-bearing when faced with a complex scenery containing conflicting cues, even when combined with lateral self-inhibition (S4 Appendix and S1A Fig).

At the computational level, the problem of HD feedback includes computing the egocentric landmark signals best suited to stabilize a drifting HD attractor. At the algorithmic level, we have proposed a solution in the form of a modified OSA. The present learning algorithm effectively depresses weights from expected visual inputs that are not consistent with the current scene, via the OSA feedback term for weight normalization. How could this feedback term be implemented? Separate regimes of synaptic depression and potentiation regimes, as well as the forward and backward transmission of information, could be orchestrated by the theta rhythm, similar to the proposed role of theta in the encoding and decoding processes in spatial memory [65–68]. That is, theta modulation of HD cells outside the generative circuitry of the attractor (e.g. in the retrosplenial cortex medial entorhinal cortex and parasubiculum [69]) could orchestrate feed-forward (at one phase of theta) vs feedback passes (at the opposite phase of theta) from visual and parietal to hippocampal regions via RSC and back, in order to associate visual inputs with the head direction system. Thus the present model may provide the first stepping stone towards a functional explanation of theta modulation in some HD cells [69], a property that is currently lacking theoretical understanding.

Note that mOSA with lateral inhibition and feedback transmission is a non-local algorithm, and could be equivalently realized by a local neural algorithm in terms of extra interneurons encoding feedback signals (e.g. [48]). Such a neural-network architecture could potentially be akin to the hippocampal CA3 back-projection onto dentate gyrus for pattern separation and storage [70,71]. Alternatively, the sparse coding in dentate gyrus may also be compatible with learning rules other than the mOSA in conjunction with autoassociative networks [71] or attractor neural networks for conflicting cue disambiguation [72].

Lateral inhibition leads to fairly narrow aLB tuning curves. We note that less sparse and more broadly tuned aLB activity could result if subsets of aLB cells do not share inhibitory connections with each other. Thus they could fire together while suppressing other aLB cells, e.g. when the strict lateral inhibition in Eq 4 is changed to other forms for Gaussian-like output firing patterns (e.g. Eq 8 in S2 Appendix). However, such broadening of aLB tuning curves would likely come at the cost of a decline in storage capacity [73–75].

## Coping with environmental changes, and 'remapping'

We have shown that aLB cells exhibit steady unimodal encoding in the presence of unstable visual features (including moving cues and teleporting cues) without any extra top-down control. If the ephemeral cue is not more salient than the remaining (stable) array of cues, the

mOSA algorithm can disconnect aLB cells from such ephemeral landmarks via long-term depression (see above discussion of the feedback-term in the learning rule) and tries to maintain synaptic connections coherent with the most recently experienced scenery. This effect offers a neural-level explanation for why stable cues tend to leave stronger fMRI signatures in RSC [53].

A novel cue can be incorporated by aLB cells, showing that blocking by previously learned features is not an issue. Absence of blocking of directional cues has been reported among HD cells [2] and place cells [54], which is likely to be explained by the absence of blocking in their upstream directional inputs (possibly from aLB cells). Moreover, even if the novel feature has been learned as part of a complex scenery, it can convey feedback (via aLB cells) on its own.

Note that, when the novel cue is more salient than the remaining array of cues, a novel set of aLB cells is recruited. This raises the question of whether or not the set of active aLB cells could provide a signal to distinguish environments at the sensory level. If only minor changes to the cue configuration are to be expected within a given environment, and more radical changes between environments, then aLB cells may help an agent distinguish between these. However, other mechanisms, such as place cell remapping [56] can also convey that information. Similarly, an agent can reasonably expect that a change of environment is accompanied by substantial path integration inputs and should not rely on sensory discrimination alone. Finally, to put the effect of cue saliency in the model into perspective, we note that within a single real-world environment with a multitude of stable features, it is very unlikely that a single added cue will overpower an already active aLB representation to recruit new cells. Hence this should only happen in novel environments, but could in principle be tested by substituting environmental (distal) cues with the cues from another familiar environment, expecting aLB remapping and a shift in the HD cell representation. Completely novel cues on the other hand, would likely only lead to the recruitment of a new set of aLB cells, and visual feedback being associated with the currently active HD cells (no HD shift), since the novel cues are not associated with strong feedback connections onto HD cells yet.

We note that the relationship between aLB cells and HD cells in the model is somewhat analogous to that suggested between place cells and grid cells. Place cells have been shown to develop links to multiple cues in a cue-rich environment [58], and their firing can subsequently be supported by a subset of cues. aLB cells are similarly associated with multiple cues (Fig 3) and remap (i.e. to a different set of active aLB cells; S3C, S4, and S5 Figs) after substantial changes to the cue array (cf. place cell remapping [56]). HD cells, on the other hand, are 're-used' throughout all environments (like grid cells), except for a coherent offset (angular for HD cells, spatial for grid cells, see [76]). HD cells implement angular path integration, whilst grid cells implement translational path integration. The sensory and path integration estimates are then combined [21] by associating aLB cells and HD cells analogous to the association of place and grid cells. That is, in the 2-dimensional spatial domain, grid cells have been proposed to update (or even generate) place cell responses [77,78], while place cells may also contribute to grid cells firing [47,79,80] and provide location feedback for grid cells (performing inherently noisy path integration) when an animal is far from environmental boundaries.

## Coding capacity

To enable spatial navigation with a large spatial scale, the navigational system with internal heading representations should be able to cope with numerous rich environments concurrently. We have shown how aLB cells and the mOSA algorithm may help accomplish a large capacity for distinct environments by allowing the model to develop distinct sets of feedback connections (activating distinct sets of aLB cells) to provide environment-specific feedback.

That is, upon re-entry into familiar environments (populated by a complex array of sensory cues) with a given heading, the present model will retrieve the correct, environment-specific HD in each environment. Capacity simulations cover both highly similar environments (demonstrating pattern separation) and more varied environments. Moreover, the capacity appears to scale, at a minimum, linearly (and possibly supra-linearly) with the number of aLB cells, with only 1080 aLB cells allowing for the correct recall of up to 16 out of 20 environments (S9C Fig). This suggests that distinct sets of feedback connections for hundreds of environments may be maintained by an aLB population with typical cortical population numbers (e.g. compared to $O(10^5)$ neurons estimated to reside in the subiculum of rodents [81]), likely outstripping any real-world need for a rodent in its lifetime. These considerations strongly suggest that the present model can, in principle, cope with a large number of distinct environments.

## Compatibility with bidirectional (BD) cells

Jacob and co-workers [28] reported bidirectional (BD) HD cells in RSC. These cells exhibit a bimodal activity profile, firing in response to identical visual landmarks in mirrored, connected environments. BD cells were solely found in dRSC [28]. These data provide further evidence for dRSC as an integrator of HD signals and visual landmark information. dRSC cells in the present model correspond to within-compartment BD (WC-BD) cells that preserve the bimodal activity profile when restricted in a single environment after exploration (similar to [30]).

Concurrently reported between-compartment BD (BC-BD) cells, seemingly solely fire for the bearing of mirrored and otherwise equal external landmarks across two rooms [28]. According to our simulations, these cells also cannot be solely explained by different, diffuse odor cues. Given that BC-BD cells keep firing in darkness [28], they are not likely to be of solely sensory origin (unlike aLB cells). Instead, BC-BD cells could occur due to inconsistent learning rates among dRSC cells that give rise to within-environment unimodality (cf. S6F Fig). However, since some BC-BD cells were reported to be spatially-modulated [28], these BC-BD cells may be derived from inhibitory control on WC-BD cells from hippocampal place cells (not part of the present model), providing a contextual signal to distinguish environments. Such inhibitory control on RSC via place cells has been proposed for parallax correction [27], providing a functional account of inhibitory connections from the hippocampus to RSC [82,83].

In the present model we have focused exclusively on landmark coding and the properties of complex environments. However, we note that making aLB cells or dRSC cells subject to spatial modulation should in principle make the present model compatible with properties of our previous model [27]. Both these models try to suggest normative explanations of HD coding in RSC: addressing the necessary computational mechanisms to cope with complex environments and location-specific feedback (e.g. for parallax correction), respectively.

## Model predictions

Beyond the replication of recent experiments, we also suggest an experimentally practical scheme to test for the presence of an aLB-like representation (Fig 7). Short of directly recording these cells (appearing as possibly narrowly tuned), HD-like cells whose responses may vanish in darkness, unless supported by other sensory modalities, such as tactile, olfactory, or acoustic cues), the conjunctive encoding of the cue array has strong implications for how conflicting cues within an array of cues act on the HD attractor. E.g. two identical cues separated by 180˚ can appear as two different rotations when considered in isolation (60˚ clockwise or 120˚ anticlockwise, red cues in Fig 7). Only by taking the relative angle between the bimodal

red and the unimodal blue cue into account can those rotation angles be distinguished. We have shown that contrary to the traditional view on the HD system (with a direct V1-RSC connection), HD cells driven by aLB cells (which treat the entire array of cues as one entity) could successfully follow such environmental rotations. Therefore, in complex environments similar to the ones used here, animals should be subject to the same shortcomings seen in the present model when aLB cells are removed.

With regard to bidirectional HD coding in RSC [28], our simulations suggest that BD-WC cells could be the product of variable learning rates across the dRSC population. Acetylcholine has been suggested to modulate learning in dRSC [82,84,85], and differential learning rates could thus be related to variations in bound neuromodulator.

We note that after removing aLB cells (i.e. treating all cues/input channels individually; Figs 7 and 8), the cortical pathway of our model resembles the snapshot model of visual feedback suggested e.g. for bees [86]. In fact, in various insects [87–90], including bees utilizing polarized skylight cues [89,91,92], the compass system (akin to HD system in rats) that provides the sense of direction with ring attractor dynamics and angular velocity integration is found in the central complex [36,93–95]. In the fly brain, such compass system in the central brain shows unimodal heading representation of the scenery with multiple salient cues [96] as well as the flexibility to adapt to multiple sensory sceneries [96,97]. Although the fast all-to-all inhibition that results in winner-takes-all dynamics on these compass neurons would induce directional ambiguation with multiple salient cues [98], this could be avoided with additional aLB cells (Figs 1 and 2). Thus, our model predicts that, if the snapshot model correctly accounts for insect data, these animals should exhibit difficulties in orienting in environments similar to the ones employed here. Similarly, the high encoding capacity shown in our model (c.f. Fig 4) could be absent.

Another prediction is that the firing of aLB cell constitutes a signal wholly dependent on sensory inputs and independent of vestibular inputs, unlike HD cells. This property could be tested in experiments that dissociate visual and vestibular signals. Finally, we have proposed that the capabilities of the model may be dependent on the theta modulation of some HD cells associating visual inputs with the HD system (see above). aLB cells also show remapping-like behavior between distinct environments, suggesting they may be 1-dimensional analogues of place cells [36,57], in stark contrast to classical HD cells.

## Conclusions

We have proposed a theoretical model of environmental stabilization of the HD system supported by abstract landmark bearing (aLB) cells. aLB cells form a compressed, sparse representation of the sensory environment. Based on a novel learning algorithm this model supports a large capacity for diverse environments, and extracts useful information across complex arrays of cues to achieve a unimodal representation of landmark bearing abstracted from any individual cue. RSC then integrates this landmark information with the HD attractor and provides feedback to stabilize HD signals. The model is robust against unstable cues, and able to incorporate new stable cues (i.e. it is resistant to blocking) without any top-down curation of landmark information. This theoretical framework integrates numerous empirical findings of various types of HD-modulated firing in RSC, and predicts a new type of cell, as well as the outcome of behavioral manipulations on the HD attractor in the presence of conflicting cues.

The present model suggests a new perspective on the neural mechanisms of spatial navigation in rich sensory environments more akin to real-world scenarios. However, like the hippocampal formation, including entorhinal cortex [5,99–101], RSC has been implicated in nonspatial coding as well [102]. With aLB cells being the product of a modified Oja's Subspace

Algorithm (which is related to PCA), an instructive avenue for future research may be to consider how non-spatial data could be processed by RSC and relayed to the medial temporal lobe, while preserving the capacity of RSC to act on the HD system. The results may suggest more general coding principles for RSC that apply to spatial and non-spatial coding alike.

## Supporting information

**S1 Appendix. Synaptic connections.**
(DOCX)

**S2 Appendix. HD attractor with a single ring.**
(DOCX)

**S3 Appendix. Feature-specific visual signals.**
(DOCX)

**S4 Appendix. Alternative algorithms.**
(DOCX)

**S5 Appendix. Simulations with three environments.**
(DOCX)

**S1 Fig. The simulation of Fig 2 with alternative algorithms.** Global representations of aLB cells via alternative algorithms after 20 minutes of learning, following a real rodent's HD trajectory. Highly activated aLB cells are sorted and labelled by positive numbers on the y-axis. The activity is ordered according to head direction (x-axis). Each row stands for a type of synaptic plasticity, while each column stands for a type of self-inhibition. Note the plot of IBCM-GI shows all aLB cells as none of them have a maximum firing rate above $\varepsilon_{aLB} = 0.5$, whilst for other plots the number of recruited aLB cells (i.e. those with firing rate above $\varepsilon_{aLB}$) varies among different algorithms (range of y-axis). Only OSA with lateral inhibition (OSA-LI) yields unimodal tuning curves for every recruited aLB cells. Warmer colors represent higher firing rates. Abbreviations: HL: (classic) Hebbian learning; HCL: Hebbian covariance learning; IBCM: Intrator's BCM; OSA: (original) Oja's Subspace Algorithm; nI: no self-inhibition; GI: global self-inhibition; LI: lateral self-inhibition; HD: head direction. Notice that OSA-nI and OSA-GI are the OSA with LI replaced by nI or GI, and mOSA is referred to as OSA-LI with non-negative weights restriction.
(PDF)

**S2 Fig. Temporal evolution of global representations of aLB cells with modified OSA.** These global representations are derived from the same testing phase and based on snapshots of the weights ($W_{\mathrm{Vis2aLB}}$) extracted every 15 seconds during the first 5 minutes of learning. The unimodal representation over a certain range of head directions has already stably emerged among aLB cells at an early stage of learning (plot with $t = 75$ $s$, 2nd row), with more recruited unimodal aLB cells representing all head directions soon thereafter (2nd, 3rd, and 4th rows). For any specific representation, all aLB cells would be shown if none of them have the maximum firing rate above $\varepsilon_{\mathrm{aLB}} = 0.5$ (global representations before $t = 60$ $s$, 1st row). Titles above each panel indicate the time (in seconds) when the snapshot of the weights was taken. Warmer colors represent higher firing rates. Abbreviations: HD: head direction.
(EPS)

**S3 Fig. Robustness with regard to ephemeral cues.** (A) The progress of synaptic weights between 'green' visual cells and aLB cells (top), with snapshots taken at the 400 s (left), 800 s (middle), and 1200 s (right), as well as the difference of these weights (bottom). The weights do

not show many changes from 800 s to 1200 s, yielding the robustness of novel cue corporation against cue removal. (B) The convergence of synaptic weights during learning. See Fig 2C for illustrations. (C) When the agent just finishes the exposure to Sc. II at 800 s, the global representation of aLB cells are tested on the 'red-blue' scenery (top), the 'red-blue-green' scenery (middle), and the 'green' scenery (bottom). These are nearly the same as the case after the whole learning in Fig 3D (i.e. tested at 1200 s), of which the details are further given in S4 Fig, yielding the robustness of aLB cells encoding against cue removal and previous scene blocking. Warmer colors represent higher firing rates of aLB cells. Abbreviations: Sc.: Scenery; HD: head direction; aLB: abstract landmark bearing; $f$: firing rate.
(PDF)

**S4 Fig. Activation patterns of aLB cells in Fig 3C.** aLB cells (x-axis with cardinal numbers irrevalant of HD) are respectively tested on the 'red-blue' scenery (Sc. I), the 'red-blue-green' scenery (Sc. II), and the 'green' scenery (Sc. II excluding the 'red-blue' scenery), when learning for 800 s (top) and 1200 s (middle), with red color stands for recruited aLB cells (i.e. $f_{aLB} \geq \varepsilon_{aLB}$ = 0.5) and white color stands for silent aLB cells (i.e. $f_{aLB} < \varepsilon_{aLB}$ = 0.5). There is little overlap between aLB cells recruited for Sc. I (1st line) and Sc. II (2nd line), suggesting the recruitment of a new set of aLB cells for Sc. II with a novel salient green cue, independent of those previously recruited for Sc. I. Some of these novel aLB cells concurrently incorporate the novel green cue (2nd and 3rd lines) without explicit learning it alone. Activation patterns of aLB cells show little changes from 800 s to 1200 s when tested on all environments (bottom; red stands for the aLB cell with changed activation), yielding the robustness against cue removal and previous scene blocking. Abbreviations: aLB: abstract landmark bearing; Sc.: Scenery.
(EPS)

**S5 Fig. Activation patterns of aLB cells in Fig 4.** aLB cells (x-axis with cardinal numbers irrevalant of HD) are respectively tested on each environment (y-axis), when learning after the first exposure (top) and at the end of learning (middle), with red color stands for recruited aLB cells (i.e. $f_{aLB} \geq \varepsilon_{aLB}$ = 0.5) and white color stands for silent aLB cells (i.e. $f_{aLB} < \varepsilon_{aLB}$ = 0.5). Little overlaps are shown among aLB cells recruited for neighbor environments with the changing green cue, suggesting the recruitment of independent sets of aLB cells. Activation patterns of aLB cells show little changes from learning the first to the last environment when tested on all environments (bottom; red stands for the aLB cell with changed activation), yielding the robustness against previous scene blocking, which further supports the high encoding capacity of aLB cells. Abbreviations: aLB: abstract landmark bearing.
(EPS)

**S6 Fig. Single-environment testing in mirrored environments with conflicting visual cues and odor cues.** (A) A pair of environments with conflicting complex sceneries (top, same as Fig 5A) and V1 signals for each visual feature (bottom, with different colors stand for corresponding features). The low-amplitude grey curve stands for the external background noise derived from a unimodal distribution over all directions. (B) Global representations of aLB and dRSC cells in S6A, tested on the scenery only in Env. I (top; same as Fig 5C, bottom) and the scenery only in Env, II (bottom), both showing bimodal firing patterns as WC-BD cells. See the caption of Fig 5 for more details. (C) A pair of environments with conflicting simple sceneries containing odors ('blue' and 'purple', independent to each other; top). Only visual signals in the second environment are shown, with odor encoding signals as uniform distributions at all directions, i.e. present for all orientations, for both two environments (bottom, here only show signals in Env. II). (D) Global representations of aLB and dRSC cells in S6C, tested on the scenery only in Env. I (top) and the scenery only in Env. II (bottom). dRSC cells still

show bimodal firing patterns when tested in any single environment, suggesting that BC-BD cells may not be solely explained by different odor cues. (E) Global representations of dRSC cells in S6C, tested on each single scenery (Env. I for top and Env, II for bottom) within darkness (i.e. unavailable 'green' cue) and with opposite initial HDs (left). The bimodal firing patterns of many dRSC cells are preserved in darkness (right), in accordance with the experimental findings. Here the higher peak of a WC-BC cell combines the HD input from gRSC with residual aLB input from the local odor cue, while the second peak is solely due to gRSC HD inputs (the aLB-odor contribution from the second environment being absent in the first, and vice versa). (F) Global representations of 360 dRSC cells in S6C, of which 180 dRSC cells learn connections slower with aLB cells and gRSC cells than other dRSC cells (S1 Table), tested on each single scenery with opposite initial HDs (left; Env. I for top and Env, II for bottom). Such partial learning among dRSC cells gives rise to both BC-BD cells (middle) and WC-BD cells (right), in accordance with the experimental findings. Abbreviations: HD: head direction; aLB: abstract landmark bearing; gRSC/dRSC: granular/dysgranular retrosplenial cortex; WC-BD: within-compartment bidirectional cell; BC-BD: between-compartment bidirectional cell; Env,: Environment; Ego.: egocentric; $f$: firing rate.
(PDF)

**S7 Fig. Three conflicting environments with simulation results in dRSC signals.** (A) Illustration of the connected environments with rotated complex sceneries (top), and the learnt connection weights form the tri-modality of dRSC responses after the whole learning phase (bottom). The agent repeatedly crosses from one environment to another, spending 60 seconds in each environment at a time, for the duration of the whole 20-minute learning phase. (B) Unimodal representations of aLB cells (top, with a single cell example) and trimodal representations of dRSC cells (bottom, with a single cell example) after learning during alternating exploration, tested in a single environment (Env. III). See the caption of Fig 5 for more details. (C) Simulation across learning shows stable HD representation. Abbreviations: HD: head direction; Diff.: difference; aLB: abstract landmark bearing; dRSC/gRSC: dysgranular/granular retrosplenial cortex; $f$: firing rate.
(EPS)

**S8 Fig. Generalized simulation results across multiple animal HD trajectory data.** aLB cells and modified Oja's subspace algorithm show identical results on simulations of Fig 2 and Fig 7 across 3 independent animal subjects (column) with their own 20-minute HD trajectory data, suggesting the generalizability of our model. Note only the data of Subject 4 (text in red) is used for simulations in all other figures. 1$^{st}$ row: Allocentric trajectories at the 11th minute of their own 20-minute HD trajectories. 2$^{nd}$ row: Global representations of sorted aLB cells showing sparsity and unimodality in Fig 2, covering the entire range of egocentric bearings despite not following any conflicting (bimodal) visual cue alone. 3$^{rd}$ row: Global unimodal stabilized representations of dRSC cells tested on Env. II in Fig 7. 4$^{th}$ row: Global representations of unimodal stabilized HD cells tested on Env. II in Fig 7. 5$^{th}$ row: HD signal representations in Fig 7 are stabilized in each environment during head-turning (blue vertical line refers to the environmental change). HD drifts across environments are in accordance with panoramic rotations rather than individual cue changes. See captions of Figs 2 and 7 for details. Abbreviations: HD: head direction; Diff.: difference; aLB: abstract landmark bearing; dRSC: dysgranular retrosplenial cortex. Rep.: representation; $f$: firing rate.
(EPS)

**S9 Fig. aLB cells show high encoding capacity across 20 rich environments apart from each other.** (A) The agent is exposed to 20 environments sequentially with feature-specific visual

input signals. In each environment, 3 randomly localized visual cues are randomly selected from 6 cues with independent features (specified in different colors). The learning duration is 2 minutes in each environment, thus 40 minutes in total, of which the HD trajectory is concatenated by two 20-minute HD trajectories used in other simulations (S1 Dataset; also see Fig 1C). (B) Global representations of aLB cells with local weights tested on corresponding sceneries. Titles for each plot refer to the corresponding environment in (A), along with the total number of highly activated aLB cells in brackets. The 'intermediate' plots provide aLB cell activity based on weights after the first exposure in each individual environment. The 'final' plots provide aLB cell activity based on weights after learning in all 20 environments is complete. See Fig 2D for illustrations. Warmer colors represent higher firing rates. (C) IoU similarity maps measuring the similarity of two sets of aLB cell firing patterns when tested on specific sceneries. Axes refer to environments, with 1 as the earliest. The left column provides IoU maps based on 'intermediate' weights. The middle column provides IoU map based on 'final' weights. The right column provides the IoU index (y-axis) between the first exposure and the end of learning tested on each scenery (x-axis). Green colors represent higher IoU, of which the maximum is 1. Red dashed line refers to IoU as 0.8, with 16 retrieved environments tested above it. (D) Positive correlation between the number of aLB cells and number of retrieved environments. Models are respectively trained with the number of aLB cells as 360 (left), 720 (middle), and 1080 (right, same as S9C right). Number of retrieved environments, which refers to as tested IoU above 0.8 (same as S9C), are stated in each plot title. Abbreviations: HD: head direction; Env.: Environment; Ego.: Egocentric; aLB: abstract landmark bearing; IoU: Intersection over Union; *f*: firing rate. See S1 Table for relevant changes in parameters for this simulation.
(TIFF)

**S10 Fig. aLB cells show high encoding capacity against visual noise across 10 environments in Fig 4.** (A-C) Same as Fig 4, except each visual input signal contains uniformly random noise (real-time change every time), with the intensity (i.e. 1-norm across all directions) as 5% of the original visual signal intensity. At least 90% of the recruited aLB cells are preserved over all 10 environments (S10C, right), suggesting the high storage capacity of these aLB cells for long-time scenery retrieval. See corresponding captions in Fig 4 for details.
(TIFF)

**S11 Fig. aLB cells show high encoding capacity against visual noise across 20 rich environments in S9 Fig.** (A-C) Same as S9 Fig, except each visual input signal contains uniformly random noise (real-time change every time), with the intensity as 5% of the original visual signal intensity. At least 80% of the recruited aLB cells are preserved over 16 out of 20 environments (S11C, right), suggesting the high storage capacity of these aLB cells for long-term scenery retrieval. See corresponding captions in S9 Fig for details.
(TIFF)

**S12 Fig. aLB cells show high encoding capacity against visual noise across 10 rich environments (cf. Fig 4).** (A) The agent is exposed to 10 environments sequentially with feature-specific visual input signals, with 2 minutes for learning each of them (same as Fig 4A). In each environment, 3 randomly localized visual cues are randomly selected from 6 cues with independent features (specified in different colors), thus richer than Fig 4A. (B) IoU similarity maps measuring the similarity of two sets of aLB cell firing patterns when tested on specific sceneries. At least 90% of the recruited aLB cells are preserved over all 10 environments (S12B, right), suggesting the high storage capacity of these aLB cells for long-term scenery retrieval even with more diverse environmental settings (cf. Fig 4C). See corresponding captions in S9

Fig for details.
(TIFF)

**S1 Table. Hyperparameters chosen for all simulations in results.** Hyperparameters of our two-stage model (with aLB cells) are shown in the third column, and those of the alternative network (without aLB cells) in the fourth column.
(DOCX)

**S2 Table. Synaptic connections in the model.**
(DOCX)

**S1 Dataset. HD trajectory data.** MATLAB extension files with 3 independent HD trajectory data from the 20-minute real-time rat recording used for simulations. Figs 1–8, S1–S7, and S9–S12 show simulation results based on the data only with Subject 1. S8 Fig shows simulation results based on the data with all 3 subjects.
(ZIP)

## Acknowledgments

The authors thank Daniel Manson for providing real head direction trajectory data (S1 Dataset).

## Author Contributions

**Conceptualization:** Yijia Yan, Neil Burgess, Andrej Bicanski.

**Formal analysis:** Yijia Yan.

**Funding acquisition:** Neil Burgess.

**Investigation:** Yijia Yan, Andrej Bicanski.

**Methodology:** Yijia Yan, Andrej Bicanski.

**Project administration:** Yijia Yan, Andrej Bicanski.

**Software:** Yijia Yan.

**Supervision:** Andrej Bicanski.

**Validation:** Yijia Yan.

**Visualization:** Yijia Yan.

**Writing – original draft:** Yijia Yan, Neil Burgess, Andrej Bicanski.

**Writing – review & editing:** Yijia Yan, Neil Burgess, Andrej Bicanski.

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
