## [Decision Letter · Decision Letter 0]

17 May 2021

Dear Yan,

Thank you very much for submitting your manuscript "A model of head direction and landmark coding in complex environments" for consideration at PLOS Computational Biology.

As with all papers reviewed by the journal, your manuscript was reviewed by members of the editorial board and by several independent reviewers. In light of the reviews (below this email), we would like to invite the resubmission of a significantly-revised version that takes into account the reviewers' comments. Specifically:

1) Both reviewers mention that the method used to achieve sparse coding may be unrealistic or too extreme. The potential effects on the model of alternative sparse encoding should be considered.

2) A more thorough justification of claims about the capacity of the network is needed, e.g., how this is affected by noise.

3) Reviewer #2's comment that the 'modified' Oja rule can effectively be reduced to the original rule by rescaling needs to be satisfactorily addressed.

The additional major and minor comments in the reviews should also be addressed.

We cannot make any decision about publication until we have seen the revised manuscript and your response to the reviewers' comments. Your revised manuscript is also likely to be sent to reviewers for further evaluation.

Sincerely,

Barbara Webb

Associate Editor

PLOS Computational Biology

Thomas Serre

Deputy Editor

PLOS Computational Biology

Reviewer's Responses to Questions

**Comments to the Authors:**

Reviewer #1: the review is uploaded as an attachment

Reviewer #2: In this manuscript by Yan et al, the authors describe a model of head direction (HD) cells and focus, in particular, on the visual input to the HD system. They propose a new hypothetical population of cells (the abstract landmark bearing cells or aLB) that improve the robustness of the HD signal and that would work as an intermediate between the visual system and the retrosplenial cells. Their input synapses would undergo synaptic plasticity via a modified Oja rule, and the neurons would be interconnected by lateral inhibition.

The paper presents, in detail, the properties of this HD system and confront it to some recent observations of non-classical tuning of retrosplenial cells. The authors also provide interesting predictions that would test for the presence of aLB cells.

Overall, the manuscript is very well written, and the analysis is thorough, presented in a logical order with sound conclusions. The proposal for the presence of aLB cells makes sense as deep layers of the visual system are thought to contain neurons with non-linear responses similar to aLB cells. To my knowledge, this kind of cell response has not been considered before in the context of HD cells, and their description deserves publication.

However, I have a few points that should be considered before publication:

## Major points

The authors modify the Oja rule on three points (l. 303), the third being a rescaling of the feedback term. It seems the one can always rescale W_{V2aLB} and \\eta in eq. 7 so that the learning dynamics is equivalent to a dynamics where \\xi =1 as in the original Oja rule (notice that W_V2aLB in equation 6 appears with a prefactor g_V2aLB that can be adjusted accordingly). The modifications that the authors propose on the Oja rule do not seem to be fundamental.

The learning rule on the aLB population leads to a winner take all dynamics whereby only a single aLB cell is active at any given time. This seems somewhat artificial. I wonder whether it would be possible to couple the aLB cells in a different way to obtain a sparse activity without the extreme sparsity that the authors describe.

The authors claim that the network has a high capacity (l 594) and that it scales with the number of aLB cells l 991. Could the author make that claim more precise? The capacity was first defined in Hopfield networks, where the capacity scales linearly with the number of neurons. In the present manuscript, the situation is different as thermal noise is not present. I think the author should explore the robustness of their learning to input noise or better define the capacity.

It is observed experimentally that both between compartment bidirectional cells (BC-BD) and within compartment bidirectional (WC-BD) cells fire in the dark. It seems that the recurrence in the present model would account for firing in the dark, but I didn’t find it claimed anywhere. Could the authors comment on that?

Also, BC-BD cells are not captured by the current model, and the authors suggest in the discussion that they could arise from a slower learning rate. Could the authors consider a heterogeneous learning rate and try to see whether BC-BD cells would appear?

## Minor points

Many of the plots are challenging to read. Most of the time, the axes are not labeled, and titles are missing.

In fig. 4B, The y axis and the order of cells should be different (they seem individually reordered). I am curious to see whether there is a correlation on the order between the different environments (even if the Jaccard index is almost zero between environments). In Fig. 4C (left and middle), I understand that we should see a 10x10 matrix visualization, but we see an intermediate green color that seems misleading.

In the discussion, the authors make the connection of their study with the insect central complex. Many observations have been made, in recent years, on the activity and learning of the drosophila central complex. This is especially true in the context of salient visual stimuli which are relevant for the discussion. The authors might want to explore this literature and see whether it is relevant for their study.

**Have the authors made all data and (if applicable) computational code underlying the findings in their manuscript fully available?**

Reviewer #1: None

Reviewer #2: **No: **I haven't seen whether the other have put their code on a public repository. It does not seem to be available either for the review.

PLOS authors have the option to publish the peer review history of their article (what does this mean?). If published, this will include your full peer review and any attached files.

Reviewer #1: **Yes: **Kathryn Hedrick

Reviewer #2: **Yes: **Hervé Rouault
---

## [Decision Letter · Decision Letter 1]

8 Sep 2021

Dear Yan,

We are pleased to inform you that your manuscript 'A model of head direction and landmark coding in complex environments' has been provisionally accepted for publication in PLOS Computational Biology.

We apologise for the delay in handling, as we were waiting on one of the original reviewers to return their review of the revised manuscript, but as this has still not occurred, we have decided to proceed.

Best regards,

Barbara Webb

Associate Editor

PLOS Computational Biology

Thomas Serre

Deputy Editor

PLOS Computational Biology

Reviewer's Responses to Questions

**Comments to the Authors:**

Reviewer #1: The reviewers thoroughly addressed all concerns. This is an interesting, well-written study that I recommend for publication.

**Have the authors made all data and (if applicable) computational code underlying the findings in their manuscript fully available?**

Reviewer #1: None

PLOS authors have the option to publish the peer review history of their article (what does this mean?). If published, this will include your full peer review and any attached files.

Reviewer #1: **Yes: **Kathryn Hedrick

---

## [Editor Report · Acceptance letter]

20 Sep 2021

PCOMPBIOL-D-21-00347R1 

A model of head direction and landmark coding in complex environments

Dear Dr Yan,

I am pleased to inform you that your manuscript has been formally accepted for publication in PLOS Computational Biology. Your manuscript is now with our production department and you will be notified of the publication date in due course.

With kind regards,

Agnes Pap
